# Improved Convex Decomposition with Ensembling and Boolean Primitives

## Abstract

Describing a scene in terms of primitives – geometrically simple shapes that offer a parsimonious but accurate abstraction of structure – is an established and difficult fitting problem. Different scenes require different numbers of primitives, and these primitives interact strongly; however, any proposed solution can be evaluated at inference time. The state of the art method involves a learned regression procedure to predict a start point consisting of a fixed number of primitives, followed by a descent method to refine the geometry and remove redundant primitives. Methods are evaluated by accuracy in depth and normal prediction and in scene segmentation. This paper shows that very significant improvements in accuracy can be obtained by (a) incorporating a small number of *negative* primitives and (b) ensembling over a number of different regression procedures. Ensembling is by refining each predicted start point, then choosing the best by fitting loss. Extensive experiments on the standard NYUv2 dataset confirm that negative primitives are useful, and that our refine-then-choose strategy outperforms choose-then-refine, confirming that the fitting problem is very difficult. Our ensembling with boolean primitives approach strongly outperforms existing methods; additionally we present several improvements to the underlying primitive generation process enabling us to obtain better decompositions with fewer primitives. Code will be released upon acceptance of the paper.

## 1 Introduction

Geometric representations of scenes and objects as *primitives* – simple geometries that expose structure while suppressing detail – should allow simpler, more general reasoning. It is easier to plan moving a cuboid through stylized free space than moving a specific chair through a particular living room. As another example, an effective primitive representation should simplify selecting and manipulating objects in scenes as in image-based scene editing (Bhat et al., 2023; Vavilala et al., 2023). But obtaining primitive representations that abstract usefully and accurately has been hard (review Sec. 2).

There are two main types of method. A **descent method** chooses primitives for a given geometry by minimizing a cost function. Important obstacles include: different geometries require different numbers of primitives; the choice of primitive appears to be important in ways that are opaque; the fitting problem has large numbers of local minima; and finding a good start point is difficult. In particular, incremental fitting procedures are often defeated by interactions between primitives. A **regression method** uses a learned predictor to map geometry to primitives and their parameters. These methods can pool examples to avoid local minima, but may not get the best prediction for a given input.

The SOTA method (Vavilala & Forsyth, 2023) for parsing indoor scenes uses a regression method to predict a start point consisting of a fixed set of primitives. An important feature of this class of problem is that, *at run time*, one can evaluate a predicted solution efficiently and accurately. The start point is polished using a descent method on a fitting loss, comparing the prediction with depth and segmentation maps from a suitable pretrained network, with backward selection to remove redundant primitives. Finally, evaluation is by comparing the primitive geometry to reference depth, normal and segmentation.

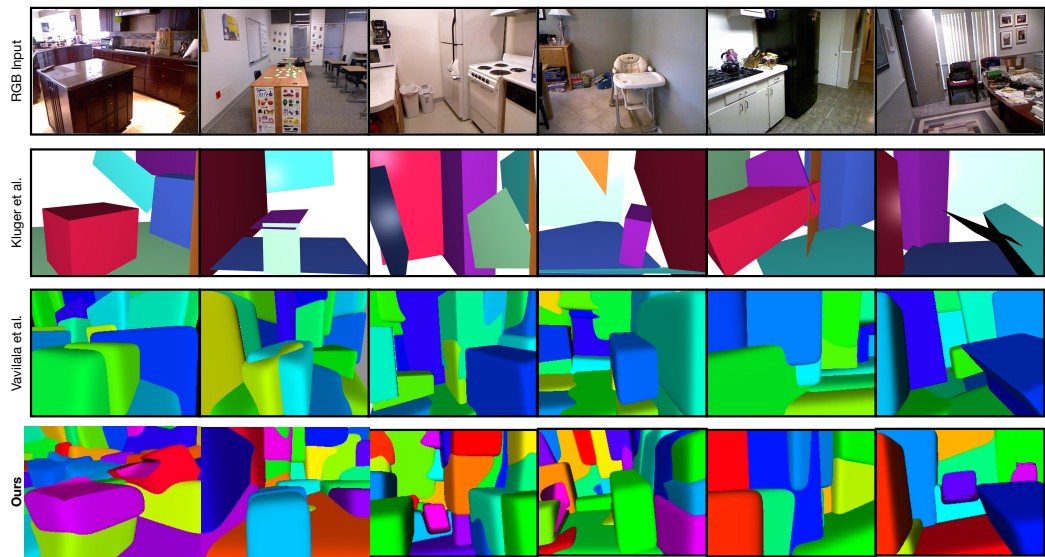

Figure 1: We present a method that advances the SOTA for primitive decomposition of indoor scenes by using ensembling and boolean primitives. We present qualitative comparison with prior work here. (**Bottom row**) In the fourth column, notice how a negative primitive helps explain free space on the bottom left; in the last column, notice how a negative primitive helps represent the chair in the center.

This paper shows two procedures that yield significant (over 40% relative error) improvements in accuracy. First, we allow a small number of *negative* primitives in the sense of constructive solid geometry (CSG). Second, we show that an appropriately constructed ensembling method produces very strong improvements in accuracy.

For **negative primitives**, the predicted geometry is the set difference between the union of positive primitives and the union of negative primitives. As our ablation experiments show, this significantly expands the geometries we can encode and significantly complicates the fitting problem. On their own, negative primitives produce small improvements in accuracy. With **ensembling** we obtain significant improvements in accuracy. We ensemble by using multiple predictors, each trained to predict a start point with a different number of primitives; some predictors use only positive primitives, others use both positive and negative primitives. Each predicted start point is then polished by minimizing a fitting loss, and the best resulting set of primitives by fitting loss is reported. This polish-then-choose strategy yields very strong improvements in accuracy. Notably, for some scenes only positive primitives are used, whereas for others both positive and negative primitives are used.

Our contributions are:

1. We believe our method is the only one that can fit CSG with a set differencing operator to indoor scenes.

2. Our novel ensembling method results in large improvements in accuracy and allows the user to control the level of abstraction. We are unaware of another method using ensembling to improve primitive generation.

3. Our primitive decomposition method for indoor scenes is an effective procedure that substantially outperforms SOTA on established metrics on the benchmark NYUv2 dataset.

## 2 RELATED WORK

Primitives date to the origins of computer vision. Roberts worked with blocks (Roberts, 1963); Binford with generalized cylinders (Binford, 1971); Biederman with geons (Biederman, 1987). Ideally, complex objects might be handled with simple primitives (Chen et al., 2019) where each primitive is a semantic part (Biederman, 1987; Binford, 1971; van den Hengel et al., 2015). Primitives

can be recovered from image data (Nevatia & Binford, 1977; Shafer & Kanade, 1983), and allow simplified geometric reasoning (Ponce & Hebert, 1982).

For individual objects, neural methods could predict the right set of primitives by predicting solutions for test data that are "like" those that worked for training data. Tulsiani *et al.* parse 3D shapes into cuboids, trained without ground truth segmentations (Tulsiani et al., 2017). Zou *et al.* parse with a recurrent architecture (Zou et al., 2018). Liu *et al.* produce detailed reconstructions of objects in indoor scenes, but do not attempt parsimonious abstraction (Liu et al., 2022). Worryingly, 3D reconstruction networks might rely on object semantics (Tatarchenko et al., 2019). Deng *et al.* (CVXNet) represent objects as a union of convexes, again training without ground truth segmentations (Deng et al., 2020). An early variant of CVXNet can recover 3D representations of poses from single images, with reasonable parses into parts (Deng et al., 2019). Meshes can be decomposed into near convex primitives, by a form of search (Wei et al., 2022). Part decompositions have attractive editability (Hertz et al., 2022). Regression methods face some difficulty producing different numbers of primitives per scene (CVXNet uses a fixed number; (Tulsiani et al., 2017) predicts the probability a primitive is present; one also might use Gumbel softmax (Jang et al., 2017)). Primitives that have been explored include: cuboids (Calderon & Boubekeur, 2017; Gadelha et al., 2020; Mo et al., 2019; Tulsiani et al., 2017; Roberts et al., 2021; Smirnov et al., 2019; Sun & Zou, 2019; Kluger et al., 2021); superquadrics (Barr, 1981; Jaklič et al., 2000; Paschalidou et al., 2019); planes (Chen et al., 2019; Liu et al., 2018a); and generalized cylinders (Nevatia & Binford, 1977; Zou et al., 2017a; Li et al., 2018). There is a recent review in (Fu et al., 2021).

Neural Parts (Paschalidou et al., 2021) decomposes an object given by an image into a set of non-convex shapes. CAPRI-Net (Yu et al., 2022) decomposes 3D objects given as point clouds or voxel grids into assemblies of quadric surfaces. DeepCAD (Wu et al., 2021) decomposes an object into a sequence of commands describing a CAD model, but requires appropriately annotated data for training. Point2Cyl (Uy et al., 2022) is similar, but predicts the 2D shapes in form of an SDF. Notably, Yu et al. (2022); Wu et al. (2021); Uy et al. (2022) also utilise CSG with negative primitives or parts but, unlike our work, focus on CAD models of single objects instead of complex real-world scenes.

Hoiem *et al* parse outdoor scenes into vertical and horizontal surfaces (Hoiem et al., 2005; 2007); Gupta *et al* demonstrate a parse into blocks (Gupta et al., 2010). Indoor scenes can be parsed into: a cuboid (Hedau et al., 2009; Vavilala & Forsyth, 2023); beds and some furniture as boxes (Hedau et al., 2010); free space (Hedau et al., 2012); and plane layouts (Stekovic et al., 2020; Liu et al., 2018b). If RGBD is available, one can recover layout in detail (Zou et al., 2017b). Patch-like primitives can be imputed from data (Fouhey et al., 2013). Jiang demonstrates parsing RGBD images into primitives by solving a 0-1 quadratic program (Jiang, 2014). Like that work, we evaluate segmentation by primitives (see Jiang (2014), p. 12), but we use original NYUv2 labels instead of the drastically simplified ones in the prior work. Also, our primitives are truly convex. Monnier *et al* and Alaniz *et al* decompose scenes into sets of superquadrics using differentiable rendering, which requires calibrated multi-view images as input (Monnier et al., 2023; Alaniz et al., 2023). Most similar to our work is that of Kluger *et al*, who identify cuboids sequentially with a RANSAC-like greedy algorithm (Fischler & Bolles, 1981; Kluger et al., 2020; 2021; 2024; Kluger & Rosenhahn, 2024).

The success of a descent method depends critically on the start point, typically dealt with using greedy algorithms (rooted in RANSAC (Fischler & Bolles, 1981); note the prevalence of RANSAC in a recent review (Kang et al., 2020)); randomized search (Ramamonjisoa et al., 2022; Hampali et al., 2021); or multiple starts. Regression methods must minimize loss over all training data, so at inference time do not necessarily produce the best representation for the particular scene. The prediction is biased by the need to get other scenes right, too. To manage this difficulty, we use a mixed reconstruction strategy – first, predict primitives using a network, then polish using descent.

## 3 METHOD

Our work is based on the architecture and losses of Vavilala & Forsyth (2023) and maintains its basic inference procedure:

1. Predict initial convex parameters from an RGBD image via a convolutional neural network.
2. Refine the fit by directly optimizing convex parameters against the training losses.

We note that generally, GT primitive decompositions are not available and instead a variety of losses supervise the fitting process. Unlike prior primitive generation work, we employ an ensemble of networks that predict varying numbers of convexes, and select the prediction which yields the lowest error after refinement (Sec. 3.1). This allows us to abandon the pruning heuristic used by Vavilala & Forsyth (2023) to control the number of convexes for each scene. We furthermore introduce *negative* boolean primitives for scene decomposition (Sec. 3.2). As visualised in Fig. 2, boolean primitives allow for a more parsimonious description of complex geometry. An additional biasing loss, annealing schedule, data augmentation, and thorough hyperparameter search yield further accuracy gains (Sec. 3.3). Fig. 3 provides an overview of our inference pipeline.

Our method is RGBD input. Our losses require a point cloud that is extracted from the depth image via the heuristic described in Vavilala & Forsyth (2023). Our method works both when GT depth is and is not available, and we evaluate both scenarios, using MIDAS (Ranftl et al., 2022) to obtain inferred depth maps.

## 3.1 ENSEMBLING

We remark that much of the literature on primitive decomposition fits a fixed number of primitives Deng et al. (2020). Other work starts from a fixed number of primitives and removes excess primitives according to a greedy algorithm (Vavilala & Forsyth, 2023). The problem with these approaches is that it is difficult to know a priori what initial settings are best for a given test image. For example, post-training refinement could get stuck in a local minimum if the start point isn't good.

A solution we employ in this work is ensembling the prediction from multiple networks, and selecting the best one. Naturally, the more members of the ensemble, the better the final quality since we can evaluate each method independently and select the best one. Our aim is simply to show one avenue of creating a usefully rich ensemble: varying the number of positive and negative primitives. Because primitive decomposition networks typically have several primary and regularizing losses, training networks with diverse hyperparameters would be another way of generating an ensemble, though due to limited compute we do not show this sort of ensemble.

Additionally, use-cases where stochasticity is desired (analogous to image generation literature) benefit from ensembling because multiple primitive decompositions will be available for a given test image. Prior work does not propose a method to generate and select from an ensemble if a user wants diverse representations of a scene.

For a given test image, we can select the best method by running it through each network, evaluating the generated primitive depth map against an inferred or GT depth, and use the best network for subsequent refinement. In practice, we observed refine-then-choose to perform better, whereby we refine each method first then choose the one with the best error metrics. Even though this involves more compute, the quality gains are substantial (see Table 1).

## 3.2 BOOLEAN PRIMITIVES

A traditional collection of primitives is represented by an indicator function $O : \mathbb{R} \to [0, 1]$, with $O(x) = 0$ indicating free space, and $O(x) = 1$ indicating a query point $x \in \mathbb{R}^3$ is inside the volume. When introducing negative primitives, the final indicator can be composed of a CSG operation between the union of positive primitives and union of negative primitives. Let $O^+(x)$ be the indicator of positive primitives only, and $O^-(x)$ be the indicator of negative primitives only. The final indicator for our representation is simply

$$O(x) = relu(O^+(x) - O^-(x)) \tag{1}$$

Our modified representation allows re-using the existing sample loss, unique parametrization loss, and Manhattan World loss Deng et al. (2020); Vavilala & Forsyth (2023) for both $O^+(x)$ and $O^-(x)$. However, for negative primitives only, we must modify the samples on which the overlap loss, guidance loss, and localization loss are applied. During each training iteration, we select samples for which the ground truth label for a point is *outside*, $x = 0$, but the indicator function is positive, $O(x) = 1$. Thus if a negative primitive moves to such a sample, its classification will become $O(x) = 0$, matching ground truth.

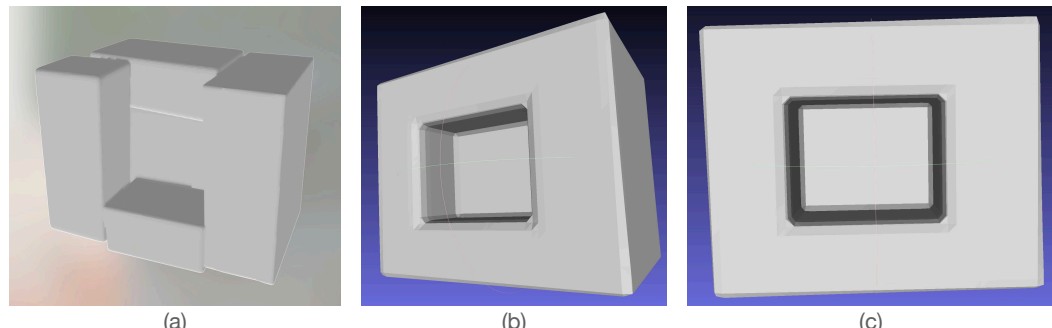

(a)         (b)         (c)

Figure 2: **Boolean primitives are parameter-efficient.** Representing a simple box with a hole punched in it can be challenging even with several traditional primitives, as shown in **(a)**, where five primitives get stuck in a local minimum. In contrast, two primitives - one positive and one negative - can represent the geometry successfully because of the enriched vocabulary of operations. Two views are shown in **(b)** and **(c)**.

Our early experimentation showed that we are better off pretraining with positive primitives only, and then introducing negative primitives for further training. Conceptually, this procedure allows positive primitives to explain the scene at a high level, and then negative primitives to improve the representation later on.

### 3.3 PERFORMANCE IMPROVEMENTS

**Biasing sample loss** The primary loss for training a convex decomposition network is

$$L_{approx} = \mathbb{E}_{x \sim \mathbb{R}^3} ||\hat{O}(x) - O(x)||^2. \tag{2}$$

We postulate that negative primitives would be most useful in regions that the positive primitives over-explain certain geometry, i.e. they explain more inside samples correctly than outside samples. In effect, if the positive primitives are "too big", then negative primitives will help the network carve away unnecessary geometry. In other words, there will be more useful regions that negative primitives can exist. We can achieve this bias by simply introducing an additional sample loss but only apply it to points where the GT label is inside, $O(x) = 1$

$$L_{inside} = \mathbb{E}_{x \sim \mathbb{R}^3} ||\hat{O}(x) - 1||^2. \tag{3}$$

We weight $L_{inside}$ by 0.1, and ablate that choice in Fig. 8.

**Annealing loss weights** Further, we found more stable training by annealing the weight of the overlap loss and alignment loss, starting from 0 at the beginning of training, up to the target weight midway through training. We preserve the annealing of the surface sample weight, whereby early in training free space samples are prioritized in the losses, and by midway of training, all samples have an equal weight. These performance improvements are intended to aid the network in predicting high-level geometric structure of the scene early in training, then getting the details right towards the end.

**Data augmentation** Prior art did not successfully implement data augmentations in the form of horizontal flips. A correct implementation needs to take into account the effect of camera calibration parameters on the point cloud. We do so here and in practice, we observe substantial improvements – see Fig. 10.

Augmentations are especially valuable given that the NYUv2 dataset is relatively small - though clearly sufficient for getting good results. Our procedure uses the standard 795/654 train/test NYUv2 split Nathan Silberman & Fergus (2012). We hold out 5% of training images for validation. We use this dataset primarily to maintain consistency in evaluating against prior art. We do not consider the volume loss or segmentation loss from Vavilala & Forsyth (2023) in our experimentation, as they were shown to have an approximately neutral effect.

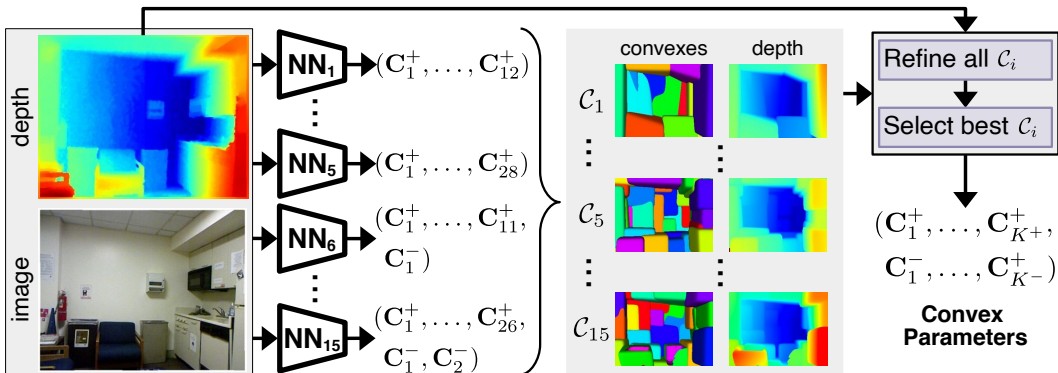

Figure 3: **Inference Overview:** We feed an RGBD image into an ensemble of independently trained convolutional neural networks. Each network predicts the parameters of a set of convexes $\mathcal{C}_i$. The number of convexes predicted by each network varies between $8$ and $40$ in this work, with up to two of them being negative. We refine each set of convexes by minimizing the training loss w.r.t. the input depth map. Our final decomposition consists of the set of refined convexes $\mathcal{C}_i$ which yields the lowest absolute relative depth error.

### 3.4 IMPLEMENTATION DETAILS

Our neural architecture is a ResNet-18 encoder (accepting RGBD input), followed by a decoder consisting of three linear layers of sizes $[1048, 1048, 2048]$ and LeakyRelu activations. We do not freeze any layers during training. The dimensionality of the final output varies based on the number of primitives the model is trained to produce (as we train different models for different numbers of primitives in this work). We implement our procedure in PyTorch and train all networks with AdamW optimizer, learning rate $4 \times 10^{-4}$, batch size 128, mixed-precision training, for 5000 iterations, on a single A40 GPU. It takes 26 mins to train a 8 primitive model and 67 mins to train a 40 primitive model. Our inference procedure requires around 5 seconds for an 8 primitive model and up to 20 seconds for a 40 primitive model. We halve the learning rate after 50% and 75% of the steps during both training and refinement. During refinement, we optimize for 250 steps, AdamW optimizer, and learning rate 0.01. Again, we find LR decay helpful during refinement with the same schedule as during training.

### 3.5 EVALUATION OF PRIMITIVES

Geometric primitive abstraction is a longstanding interest in computer vision ( Marr & Nishihara (1978), Sec. 2) but finding broad applications of them is ongoing. We are aware of recent efforts to condition image synthesis on primitives - see Vavilala et al. (2023); Bhat et al. (2023). Our work can be a useful building block for such use-cases.

To that end, we'd like to evaluate primitives appropriately for these tasks. If a user is assembling primitives (possibly extracted from a real image) and editing their position, it is critical that the generated image matches the requested depth. Thus, our evaluation must measure the geometric accuracy between the generated primitives and the source depth. If we are allowed a large number of primitives e.g. 1 per pixel, we could perfectly match the GT depth at the cost of coarse scene abstraction; because we are instead dealing with few primitives (8-40 in this work), depth error metrics will give a true indication of geometric scene decomposition quality. Similarly, evaluating per-pixel normals offers a measure of geometric reconstruction quality. GT depth and normals are available at inference time (and they can be inferred by high quality estimators like Ranftl et al. (2020) if not). Predicted depth and normals can be obtained by ray marching the generated primitives from the original viewpoint, obtaining a dense per-pixel estimate. Similarly, downstream use-cases of primitives may require object-level control. We can evaluate how well our primitives enable this capability by assigning each primitive's face the most common GT segmentation label in its support, and then measuring the dense per-pixel segmentation accuracy across the whole image. A high score means that primitives map to objects quite well.

We note that we evaluate these metrics against existing primitive generation works from RGB images (specifically Vavilala & Forsyth (2023); Kluger et al. (2021)), not against methods that predict segmentation, depth, or normals from RGB like Yang et al. (2024); Kirillov et al. (2023). We argue that we can evaluate primitives by looking at their predicted depth/normals/segmentation. But we're not trying to predict depth/normals/segmentation from RGB using primitives, for which there are well-developed methods. Our evaluation procedure is consistent with prior art. Better metrics should mean better primitives. This allows us to use detailed quantitative evaluations in an area that has traditionally lacked them.

## 4  EXPERIMENTS

We perform extensive quantitative and qualitative evaluation of our method. To do so, we use established evaluation procedures on the depth, normals, and segmentation inferred from the generated primitives.

**Any individual network we train beats baselines**. Without ensembling, with or without negative primitives, our method beats all baselines on nearly every metric - see Tables 1, 2. Here, we present several networks with different numbers of positive and negative primitives. We apply refinement at test time. Individually, each procedure performs quite well across a range of initial primitives. In some cases, introducing negative primitives helps on average (we test $K^- \in [0, 1, 2]$). When we ensemble the five networks without negative primitives, we get substantially better error metrics, particularly as measured by AbsRel of the depth map. To select the best method, we simply compare the depth of generated primitives against GT. **Ensembling with negative primitives can boost quality**. Further, when we enrich the vocabulary of operations with negative primitives, depth metrics get better (**pos+neg R->S**).

**Refinement improves all methods**. In Fig. 7, we apply our refinement procedure on all test images using the GT depth map. Consistent with previous work, refining is essential to getting the best results. Observe how all error metrics, particularly AbsRel, get better with refinement. In particular, the negative primitives we introduced get better with refinement. While we get strong results across all numbers of primitives, the introduction of negative primitives only occasionally helps on average, in some cases slightly hurting metrics, which indicates that our test scenes are quite diverse and different settings are optimal for each scene. **Refine-then-select performs better than select-then-refine**. When we ensemble the five positive-only networks, rows **pos**, all error metrics get better than any method alone. However, the fact that we get better numbers when we select after refining indicates that this is an extremely difficult fitting problem whereby what appears to be the best start point may not necessarily yield the best endpoint. When comparing ensembles with negative primitives (**pos+neg**), we again observe that we are better off refining then selecting. Further, on average the network picks 0.30 negative primitives in our best ensemble - which means they are genuinely helpful on some scenes. In Fig 4 we present histograms showing how many total and negative primitives were chosen on our test set. In practice, (**left**) our procedure is able to handle larger numbers of primitives better than prior work, observing that more primitives is generally better, and **right**, negative primitives can be quite helpful, noting that they are selected from the ensemble for several scenes.

**A biased sampling loss should be part of the ensemble**. We ablate our decision to bias the sampling loss to favor classifying "inside" points correctly via $L_{inside}$. In Fig. 8, we test $w_{inside} \in [0.0, 0.1, 0.2, 0.4, 0.8]$ with $K^- \in [1, 2]$. Turning this loss on is generally preferred over not having it. We thus let $w_{inside} = 0.1$ in our experimentation as a reasonable mid-ground for networks with negative primitives.

**Data augmentation yields more accurate decompositions.** As Fig. 10 shows, augmenting RGBD input data with horizontal flips during training reduces the AbsRel depth error and increases the segmentation accuracy measurably, with more modest effects on normal accuracy.

## 5  DISCUSSION

The key goal of primitive decompositions since the 1960s has been to demonstrate representations that can (a) be computed from data and (b) genuinely simplify reasoning tasks. We have demonstrated

Table 1: We quantitatively evaluate our ensembling approach. **First five rows**: we train a primitive generation model according to the procedure laid out in Sec 3. The value under the method column indicates the number of primitives, and no negative primitives are shown here. Next four rows: ensembling strongly improves error metrics across the board (we focus on depth, normals, and segmentation accuracy). Pos refers to five networks with only positive primitives in the ensemble ($K^{total} \in [8, 16, 24, 32, 40]$); Pos+Neg refers to fifteen networks in the ensemble (where $K^- \in [0, 1, 2]$). $S \rightarrow R$ means that we evaluate a given test image on each method in the ensemble without finetuning, then finetune the best one using the original network's training losses. In this table, we finetune assuming GT depth is available at test time, though our method still works even when depth is inferred by a pretrained depth estimator. $R \rightarrow S$ means that we refine the primitives generated by each method for a given test image, then pick the best one (as measured by AbsRel). The fact that substantial gains can be achieved from $R \rightarrow S$ implies that the best start point may not yield the best end point – meaning the fitting problem is hard. Time and memory estimates are presented as well. **Last row**: we compare our methods against existing work. Any individual model we train obtains better error metrics with less compute. Timings for ensembling show estimated total cost of running all the methods and selecting the best one; memory refers to peak GPU memory usage.

| Method | Time (s) | Memory (GB) | AbsRel $\downarrow$ | Normals$_{mean}$ $\downarrow$ | Normals$_{median}$ $\downarrow$ | Seg$_{acc}$ $\uparrow$ |
|---|---|---|---|---|---|---|
| 8 | 5.23 | 2.13 | 0.095 | 37.0 | 31.7 | 0.574 |
| 16 | 9.39 | 3.76 | 0.0714 | 35.7 | 30.0 | 0.653 |
| 24 | 11.9 | 5.71 | 0.0662 | 35.3 | 29.9 | 0.678 |
| 32 | 15.9 | 7.15 | 0.0613 | 35.4 | 29.8 | 0.697 |
| 40 | 18.8 | 8.77 | 0.0645 | 35.2 | 29.7 | 0.694 |
| Pos - S$\rightarrow$R | 16.7 | 8.77 | 0.0666 | 35.6 | 30.0 | 0.666 |
| Pos + Neg S$\rightarrow$R | 25.7 | 8.77 | 0.0672 | 35.8 | 30.2 | 0.668 |
| Pos - R$\rightarrow$S | 61.3 | 8.77 | 0.0561 | **35.1** | **29.5** | **0.698** |
| Pos + Neg R$\rightarrow$S | 184 | 8.77 | **0.0545** | 35.2 | 29.6 | **0.698** |
| Vavilala 2023 | 40.0 | 6.77 | 0.0980 | 37.4 | 32.4 | 0.618 |

Table 2: **Baseline comparisons:** Ensembling strongly outperforms two recent SOTA methods, using the metrics reported by Kluger et al. (2021), and using negative primitives in the ensemble produces further improvements in some cases. We show results with only positive primitives present **Ours (pos)**, five networks, $K^{total} \in [8, 16, 24, 32, 40]$, as well as with positive and negative primitives **Ours (pos+neg)**, 15 networks, $K^- \in [0, 1, 2]$. Our ensembles significantly outperform existing work. Further, we present results on the fifteen methods we trained, where $K^{total}/K^-$ is shown. Even without ensembling, any individual method we trained generally performs better than the baselines.

| Ensemble | Refine | $K^{total}$ | $K^-$ | AUC$_{@50}$$\uparrow$ | AUC$_{@20}$$\uparrow$ | AUC$_{@10}$$\uparrow$ | AUC$_{@5}$$\uparrow$ | mean$_{cm}$$\downarrow$ | median$_{cm}$$\downarrow$ |
|---|---|---|---|---|---|---|---|---|---|
| No (Vavilala 2023) | Yes | 13.9 | 0 | 0.869 | 0.725 | 0.565 | 0.382 | 0.266 | 0.101 |
| No (Kluger 2021) | N/A | - | 0 | 0.772 | 0.627 | 0.491 | 0.343 | 0.208 | - |
| No | Yes | 8 | 0 | 0.8728 | 0.7521 | 0.6098 | 0.4378 | 0.2547 | 0.0837 |
| No | Yes | 8 | 1 | 0.8558 | 0.7401 | 0.6024 | 0.4297 | 0.2863 | 0.0888 |
| No | Yes | 8 | 2 | 0.8584 | 0.7419 | 0.6049 | 0.4350 | 0.2815 | 0.0860 |
| No | Yes | 16 | 0 | 0.9092 | 0.8297 | 0.7173 | 0.5513 | 0.1920 | 0.0548 |
| No | Yes | 16 | 1 | 0.8888 | 0.8038 | 0.6890 | 0.5218 | 0.2258 | 0.0609 |
| No | Yes | 16 | 2 | 0.8881 | 0.8043 | 0.6902 | 0.5210 | 0.2317 | 0.0616 |
| No | Yes | 24 | 0 | 0.9133 | 0.8420 | 0.7346 | 0.5698 | 0.1855 | 0.0512 |
| No | Yes | 24 | 1 | 0.8930 | 0.8183 | 0.7120 | 0.5500 | 0.2167 | 0.0547 |
| No | Yes | 24 | 2 | 0.8943 | 0.8159 | 0.7042 | 0.5348 | 0.2169 | 0.0595 |
| No | Yes | 32 | 0 | 0.9177 | 0.8546 | 0.7576 | 0.6006 | 0.1755 | 0.0458 |
| No | Yes | 32 | 1 | 0.8782 | 0.8051 | 0.7067 | 0.5534 | 0.2415 | 0.0573 |
| No | Yes | 32 | 2 | 0.8904 | 0.8201 | 0.7184 | 0.5562 | 0.2267 | 0.0534 |
| No | Yes | 40 | 0 | 0.9113 | 0.8487 | 0.7503 | 0.5918 | 0.1869 | 0.0486 |
| No | Yes | 40 | 1 | 0.8936 | 0.8258 | 0.7335 | 0.5818 | 0.2141 | 0.0500 |
| No | Yes | 40 | 2 | 0.8903 | 0.8229 | 0.7276 | 0.5729 | 0.2196 | 0.0528 |
| pos | S->R | 23.3 | 0 | 0.9124 | 0.8353 | 0.7262 | 0.5628 | 0.1852 | 0.0530 |
| pos + neg | S->R | 24.5 | 0.5 | 0.9059 | 0.8288 | 0.7211 | 0.5594 | 0.1961 | 0.0539 |
| pos | R->S | 31.8 | 0 | 0.9259 | **0.8617** | **0.7617** | **0.6017** | 0.1612 | **0.0456** |
| pos + neg | R->S | 31.7 | 0.3 | **0.9265** | 0.8616 | 0.7614 | 0.6010 | **0.1603** | 0.0457 |

a method that can produce accurate fits of multiple convex primitives, some "negative," to complex indoor scenes represented in RGBD images. Our method really can be computed from data, and in accuracy significantly outperforms SOTA.

**Limitations** The method requires ensembling a number of regressors, with consequent costs in training and inference time. While we can evaluate accuracy, it is difficult to usefully assess the extent to which the method is parsimonious, apart from looking at the relatively small number of primitives used. We have shown partial progress on simplifying reasoning tasks (the depth implied by the primitives is quite good, and the segmentation is fair but not competitive with the best semantic segmenters). In this work we selected a modest network size and small benchmark dataset (to temper compute requirements and perform evaluation); scaling the model architecture and dataset is a natural extension.

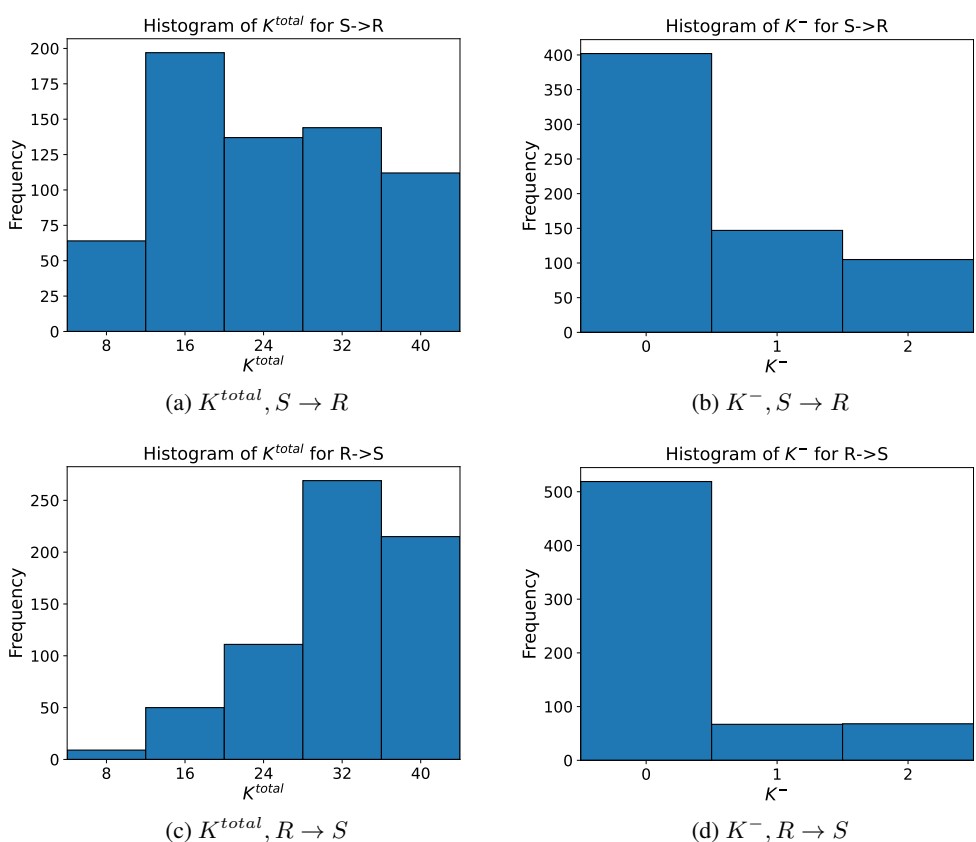

(a) $K^{total}, S \rightarrow R$

(b) $K^{-}, S \rightarrow R$

(c) $K^{total}, R \rightarrow S$

(d) $K^{-}, R \rightarrow S$

Figure 4: We analyze our ensembling procedure by breaking down which models are ultimately chosen when selecting then refining (**top row**) or refining then selecting (**bottom row**). When selecting then refining, all primitive counts are well represented in the ensemble, with 16 slightly preferred. When refining then selecting, the model strongly favors more primitives, whereby 32 is the most commonly picked. Interestingly, some scenes prefer fewer primitives, which can be due to fitting difficulties for a particular test image with larger numbers of primitives. While one would expect more primitives to lead to better quality, we observe a drop-off in quality around 32 primitives, noting that 40 is chosen less often than 32. This could be due to bias-variance issues in the network and challenges in optimizing larger numbers of primitives. (**right column**) Our method generally prefers not using negative primitives, but occasionally selects them, indicating they are genuinely useful in some scenes.

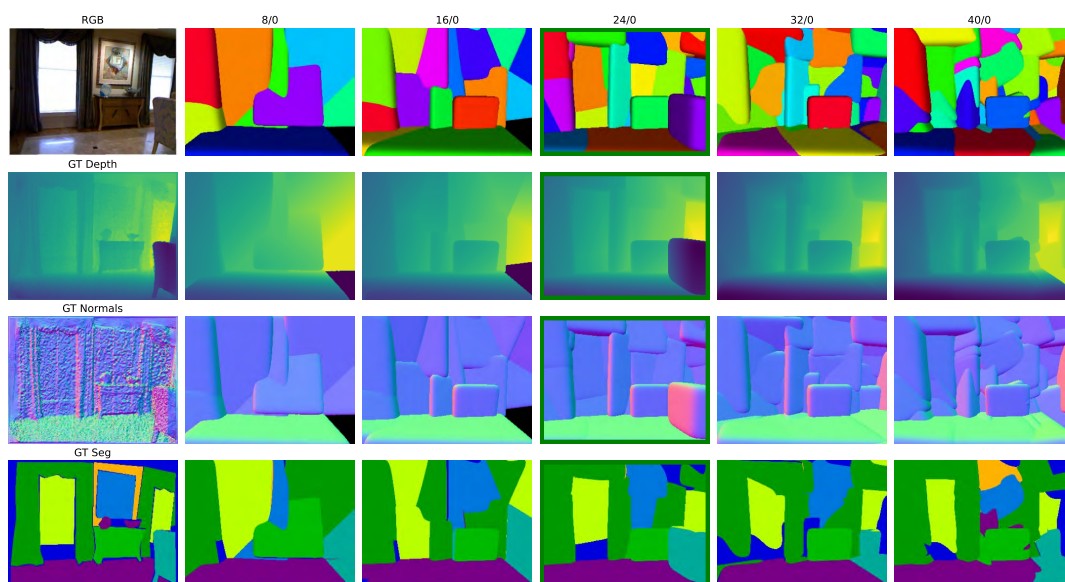

Figure 5: We present qualitative evaluation of our ensembling procedure. The first column shows GT information, including the RGB input and GT Depth map accepted by the model. The remaining columns show generated results with $K^{total}/K^{-}$ shown in the first row. The model chosen by ensembling (comparing AbsRel of the depth from primitives against GT depth) is boxed in green. Depth/normals from primitives is obtained by ray-marching from the original camera view; predicted segmentations are obtained by assigning each primitive's face the most common GT label within its support.

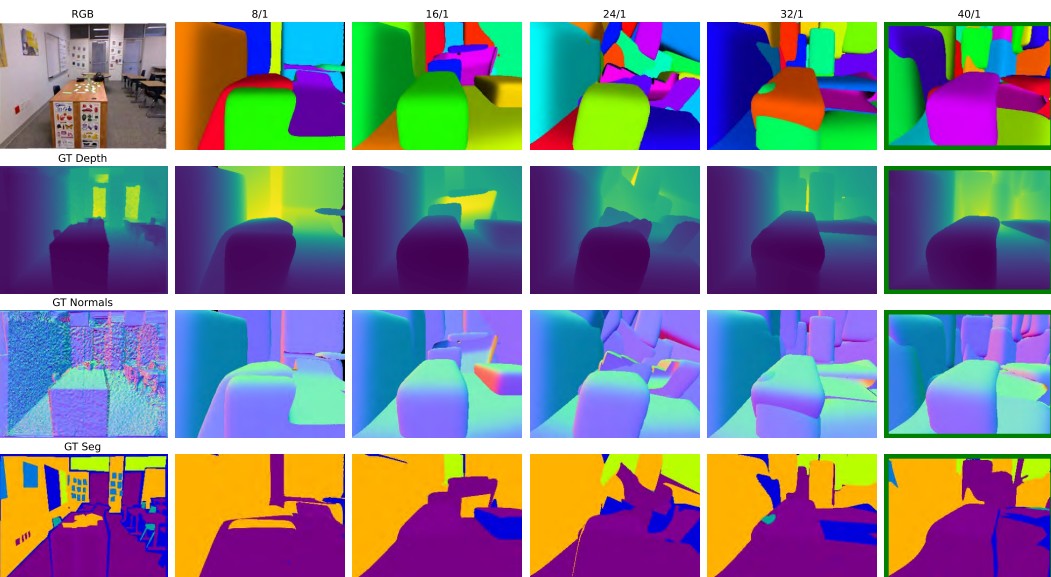

Figure 6: Additional qualitative evaluation with negative primitives. In this case, 40 primitives (with 1 negative primitive) were chosen. The negative primitive in $40/1$ was placed in the bottom right of the image to indicate free space.

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

# A    APPENDIX

Table 3: We present metrics without finetuning, and with GT depth available at test time. Notice how metrics are much worse without post-training refinement.

| Ensemble | Refine | $K^{total}$ | $K^-$ | AbsRel $\downarrow$ | Normals$_{mean}$ $\downarrow$ | Normals$_{median}$ $\downarrow$ | Seg$_{acc}$ $\uparrow$ |
|---|---|---|---|---|---|---|---|
| No | No | 8 | 0 | 0.1669 | 40.4625 | 37.3284 | 0.5370 |
| No | No | 8 | 1 | 0.1862 | 41.4633 | 38.0116 | 0.5328 |
| No | No | 8 | 2 | 0.1872 | 42.5348 | 39.3306 | 0.5335 |
| No | No | 16 | 0 | 0.1547 | 41.3356 | 37.6371 | 0.6027 |
| No | No | 16 | 1 | 0.1627 | 42.0006 | 38.5059 | 0.5963 |
| No | No | 16 | 2 | 0.1797 | 43.1610 | 39.5093 | 0.5912 |
| No | No | 24 | 0 | 0.1566 | 41.6409 | 38.4479 | 0.6264 |
| No | No | 24 | 1 | 0.1695 | 43.2274 | 39.4391 | 0.6166 |
| No | No | 24 | 2 | 0.1726 | 42.9552 | 39.7228 | 0.6103 |
| No | No | 32 | 0 | 0.1549 | 44.2891 | 40.1252 | 0.6579 |
| No | No | 32 | 1 | 0.2145 | 46.0952 | 41.7799 | 0.6026 |
| No | No | 32 | 2 | 0.1871 | 43.1564 | 39.6844 | 0.6188 |
| No | No | 40 | 0 | 0.1672 | 42.0187 | 39.0446 | 0.6524 |
| No | No | 40 | 1 | 0.1642 | 47.1960 | 42.4641 | 0.6620 |
| No | No | 40 | 2 | 0.1667 | 43.3853 | 39.8320 | 0.6304 |

Table 4: We present quantitative evaluation of the 15 models we trained, but the best strategy by far is to ensemble (bottom block). Best AbsRel was the criteria used to select a model for a given test image. Generally, refine-then-select (R→S) is significantly better than select-then-refine (S→R), likely because the fitting problem is extremely hard, so the start point for refining is not a good guide to how well the refinement will proceed. In the bottom block, the $K^-$ indicates the average number of negative primitives used per image, suggesting the best fit for a significant fraction of images has one or more negative primitives. First two rows show recent prior work. Any individual model as well as any ensemble generally outperforms prior work across all error metrics. Final row: the very best depth accuracy, as measured by AbsRel, was achieved by using an ensemble with negative primitives. Boolean primitives improved AbsRel and Segmentation accuracy on average when we use 8 primitives, but hurt the quality on average for more than 8 primitives. The implication is that fitting boolean primitives remains hard. However, the advantage of ensembling is that boolean primitives will only be used where they are helpful.

| Ensemble | Refine | $K^{total}$ | $K^-$ | AbsRel ↓ | Normals$_{mean}$ ↓ | Normals$_{median}$ ↓ | Seg$_{acc}$ ↑ |
|---|---|---|---|---|---|---|---|
| No (Vavilala 2023) | Yes | 13.9 | 0 | 0.098 | 37.355 | 32.395 | 0.618 |
| No (Vavilala 2023) | Yes | 15.7 | 0 | 0.096 | 37.355 | 32.700 | 0.630 |
| No | Yes | 8 | 0 | 0.0949 | 36.9861 | 31.7493 | 0.5741 |
| No | Yes | 8 | 1 | 0.0944 | 37.7630 | 32.4935 | 0.5743 |
| No | Yes | 8 | 2 | 0.0911 | 38.2590 | 32.7630 | 0.5774 |
| No | Yes | 16 | 0 | 0.0714 | 35.7310 | 30.0465 | 0.6525 |
| No | Yes | 16 | 1 | 0.0741 | 36.6899 | 30.8987 | 0.6455 |
| No | Yes | 16 | 2 | 0.0754 | 36.7649 | 30.9506 | 0.6456 |
| No | Yes | 24 | 0 | 0.0662 | 35.2619 | 29.8957 | 0.6776 |
| No | Yes | 24 | 1 | 0.0712 | 36.5494 | 30.8535 | 0.6642 |
| No | Yes | 24 | 2 | 0.0707 | 36.5984 | 31.3036 | 0.6653 |
| No | Yes | 32 | 0 | 0.0613 | 35.4398 | 29.7855 | 0.6970 |
| No | Yes | 32 | 1 | 0.0782 | 37.0885 | 31.4945 | 0.6721 |
| No | Yes | 32 | 2 | 0.0721 | 36.4009 | 30.8432 | 0.6742 |
| No | Yes | 40 | 0 | 0.0645 | 35.1675 | 29.7039 | 0.6942 |
| No | Yes | 40 | 1 | 0.0697 | 36.8514 | 31.3076 | 0.6942 |
| No | Yes | 40 | 2 | 0.0712 | 36.0667 | 30.4413 | 0.6832 |
| pos | S->R | 23.3 | 0 | 0.0666 | 35.5563 | 29.9633 | 0.6662 |
| pos + neg | S->R | 24.5 | 0.5 | 0.0672 | 35.8283 | 30.1908 | 0.6679 |
| pos | R->S | 31.8 | 0 | 0.0561 | **35.1100** | **29.5008** | **0.6984** |
| pos + neg | R->S | 31.7 | 0.3 | **0.0545** | 35.2119 | 29.5695 | 0.6975 |

Table 5: We ablate the choice to perform learning rate decay (halved once midway through training, again after 75% of the steps, (**LR DECAY ON**) versus leaving it at a constant value (**LR DECAY OFF**). AbsRel values shown in the table for varying numbers of total and negative primitives, on a portion of the NYUv2 test set. The results generally favor using LR decay.

| | $K^{total} = 8$ | | | $K^{total} = 24$ | | |
|---|---|---|---|---|---|---|
| $K^-$ | 0 | 1 | 2 | 0 | 1 | 2 |
| LR DECAY OFF | 0.098 | 0.099 | 0.108 | **0.067** | **0.073** | 0.081 |
| LR DECAY ON | **0.090** | **0.091** | **0.093** | **0.067** | 0.076 | **0.077** |

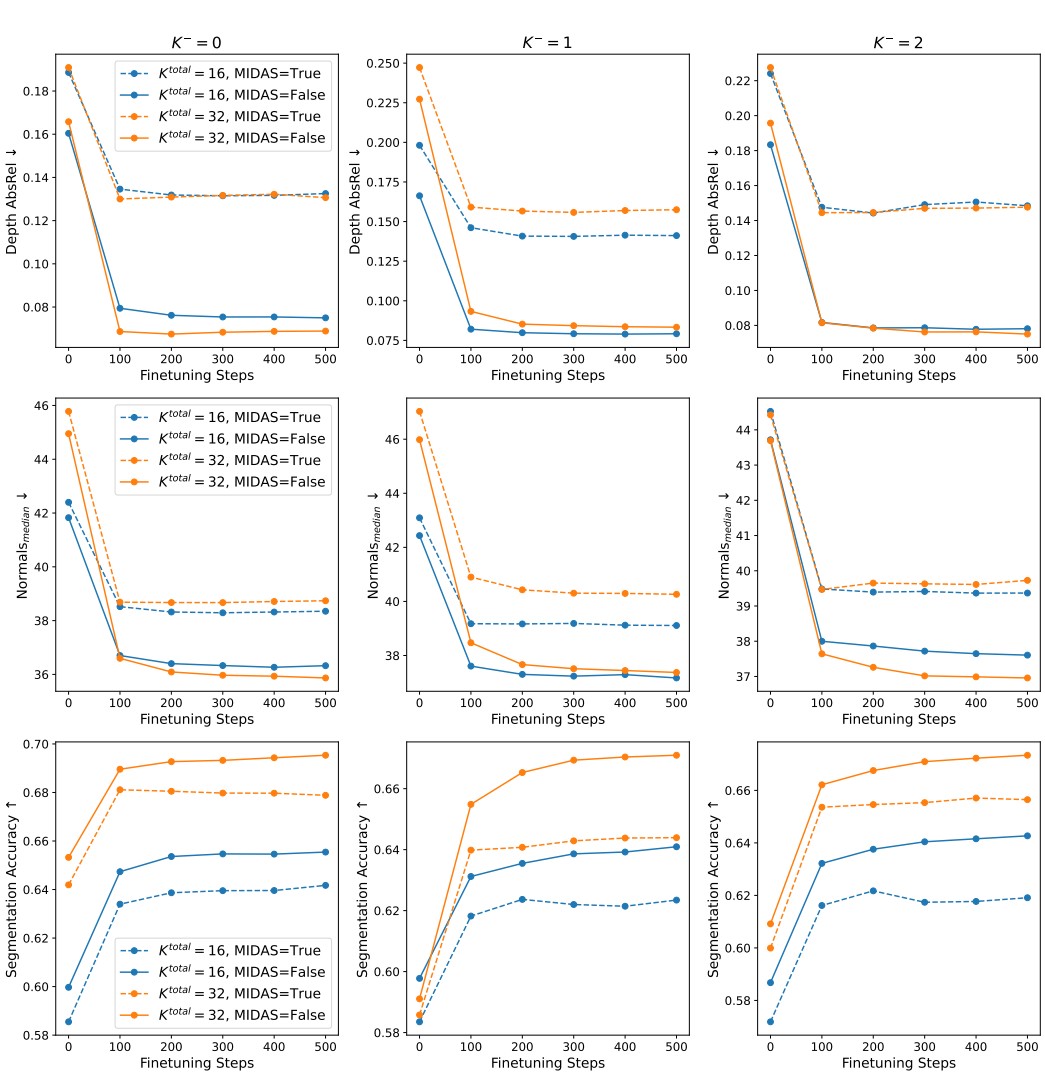

Figure 7: We demonstrate why finetuning is important for primitive generation. Running a primitive generation model alone gives reasonable start points, but note how after a small amount of finetuning, all metrics get much better. This is true across primitive counts (we show $K^{total} \in [16, 32]$ here), presence of negative primitives (a different $K^-$ shown in each column), and whether GT depth is available at test time (MIDAS = True) or not (MIDAS = False). To perform test-time refinement, we directly optimize the parameters of the primitives with respect to the training losses. In this work, we use 250 refinement steps per test image, a reasonable balance between speed and quality. We note that previous work has established that refining from a random start point does not yield good results (Vavilala & Forsyth, 2023).

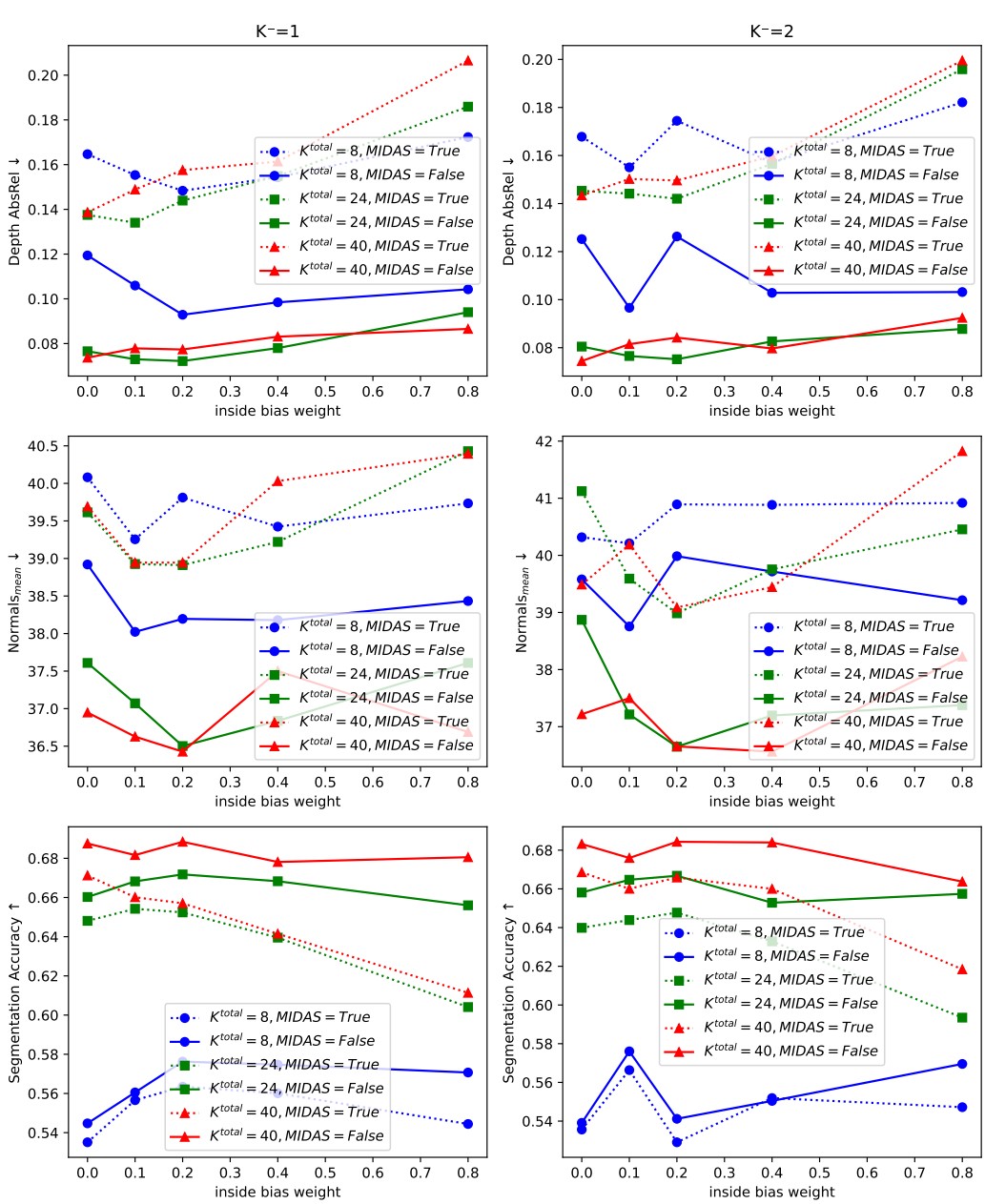

Figure 8: We ablate choices for our biased loss term in Equation 3, which only applies when negative primitives are present. Varying numbers of primitives, are shown with different colors and tick labels, and regimes where GT depth is and is not available at test time are shown. Each row shows a different error metric, and each column shows a different number of negative primitives. Overall, it appears having a small amount of this bias term is advantageous.

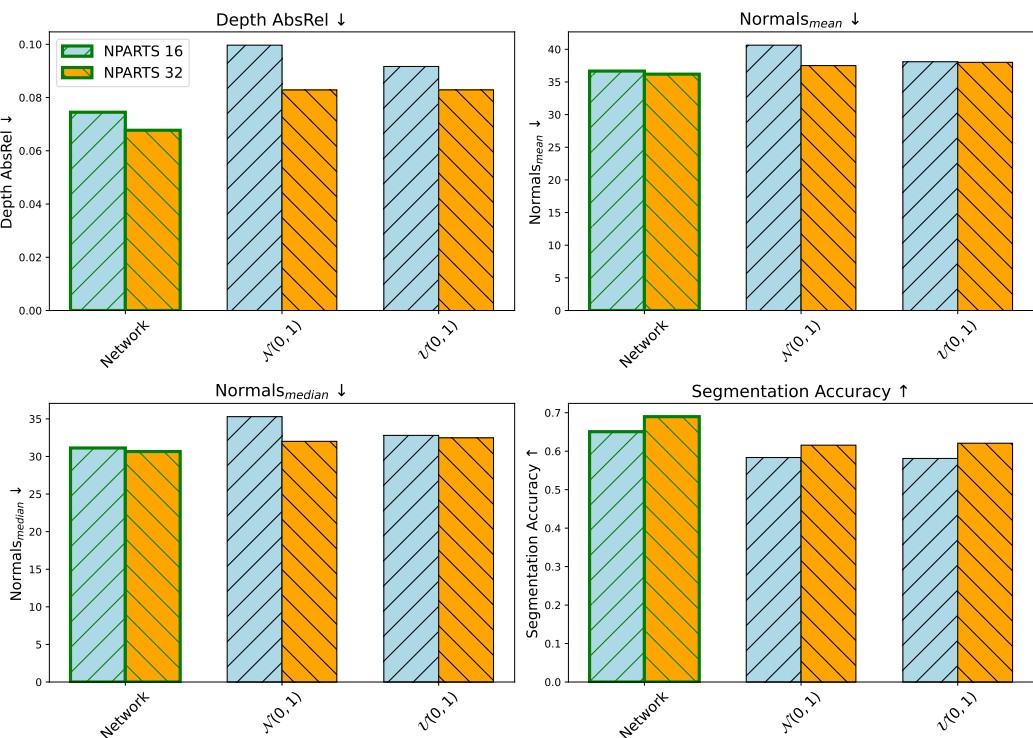

Figure 9: We demonstrate that initializing our refinement process with primitives predicted by a network is advantageous. For $K^{total} \in [16, 32]$, all metrics are better with network start, as opposed to fitting with randomly initialized parameters (we show both normal and uniformly distributed initializations). We allow each method to optimize for a very long time (3000 steps). One line of future work could be better initialization that avoids the need to train a neural network, for example initializing primitives near centers obtained by another method (like Wei et al. (2022)). Another line of work could be improving the network start by scaling the network and dataset to potentially reduce the need for refinement.

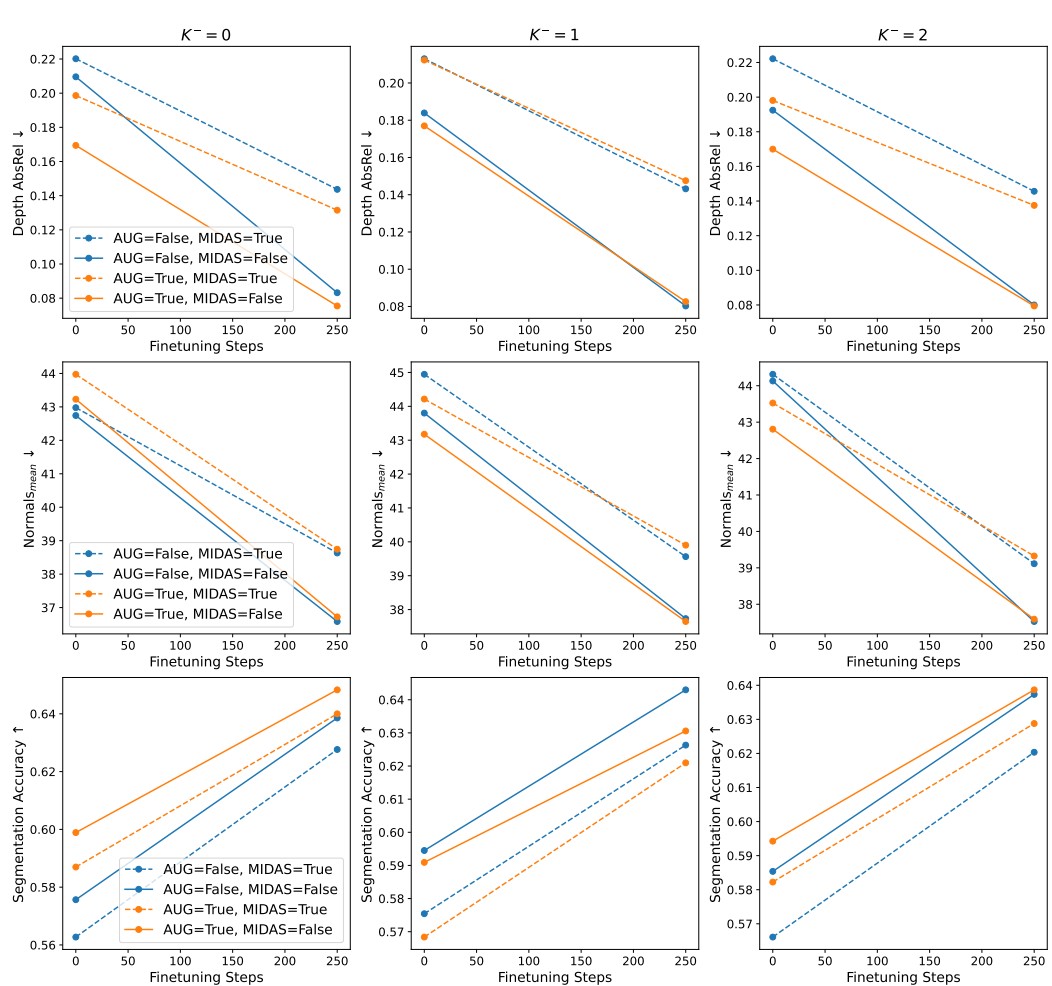

Figure 10: Introducing X-flip augmentations during training generally improves error metrics. We test this on $K^{total} = 16$

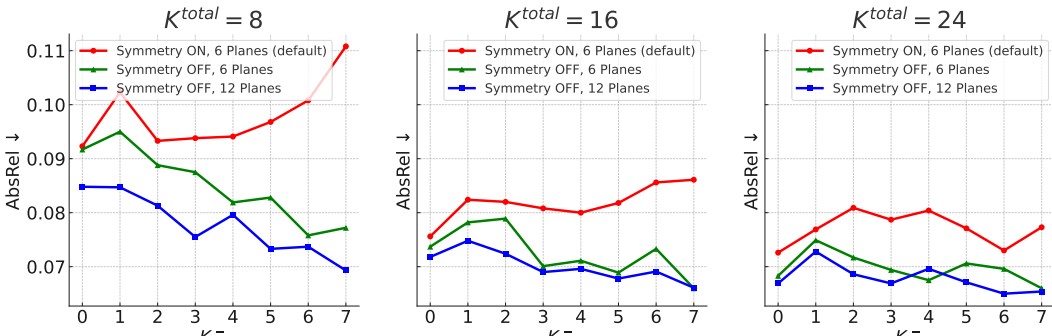

Figure 11: We perform an ablation on the number of negative primitives, $K^-$ as well as the primitive vocabulary. By default, in this work we generate parallelepipeds (a more general form of a cuboid) to maintain consistency in evaluation against prior work Vavilala & Forsyth (2023); Kluger et al. (2021). To do so, our model predicts three normals and offsets per primitive, and the other three are implied. Thus the primitives are centrally symmetric. Our experimentation shows that fitting CSG with parallelepipeds is very difficult, as indicated by the AbsRel getting worse as we increase the number of boolean primitives (red line). However downstream use-cases may not require the centrality constraint and good reconstruction quality might be paramount. To that end, we try two more ablations. First, we remove the centrality constraint and Manhattan World loss (green line). Notice how all numbers get better and in particular primitive decompositions get better with more boolean primitives. We then increase the number of halfplanes to 12, (blue line) and the quality is generally better across the board. The implication is that fitting CSG is easier if we fit primitives with a more flexible parametrization (convex polytopes) as opposed to more rigid primitives (e.g. cuboids). We remark that within each subplot, the total number of primitives remains the same ($K^{total}$) and we are simply adjusting the ratio of positive and negative primitives ($K^+/K^-$). Our implementation supports this richer primitive vocabulary by simply tuning hyperparameters. Experiments conducted on a portion of the NYUv2 test set.

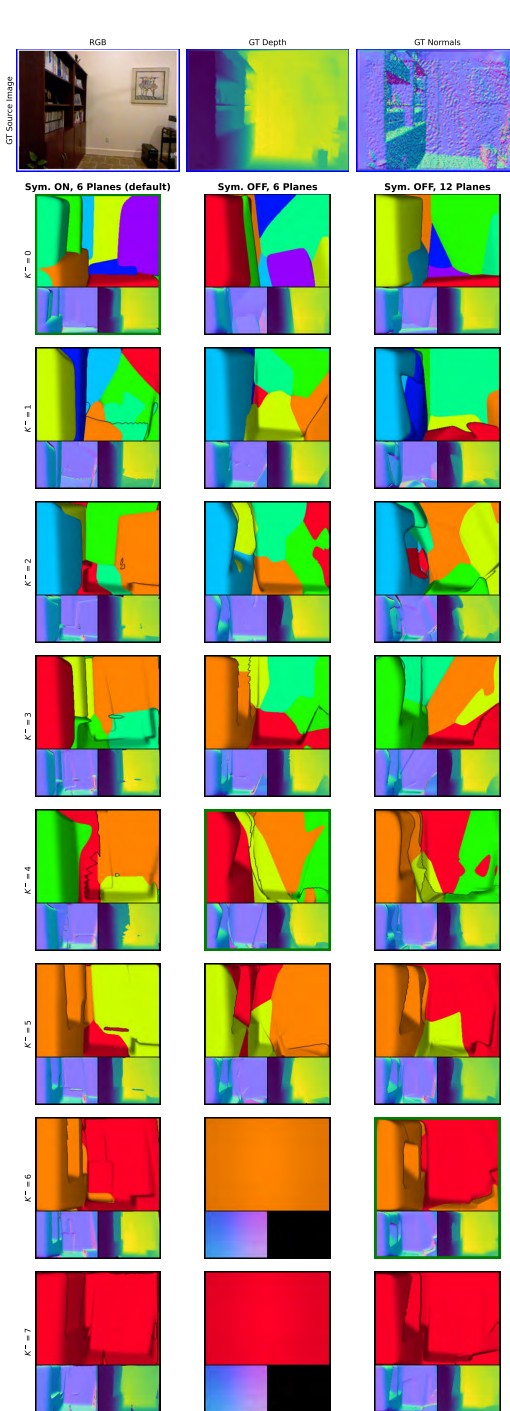

Figure 12: We perform a qualitative evaluation on the number of boolean primitives, $K^- \in [0, 1, ...7]$, with all images having the same $K^{total} = 8$. In each column, the decomposition with lowest AbsRel selected by ensembling is boxed in green. We decompose parallelepipeds with a Manhattan World constraint (**first column**), general 6-face polytopes (**second column**), and 12-face polytopes (**third column**). Notice how boolean primitives carve away free space in the bookshelf on the left side of each image. The final two entries of the middle column reached a degenerate state during the optimization process and failed to recover, which further justifies the benefits of ensembling.

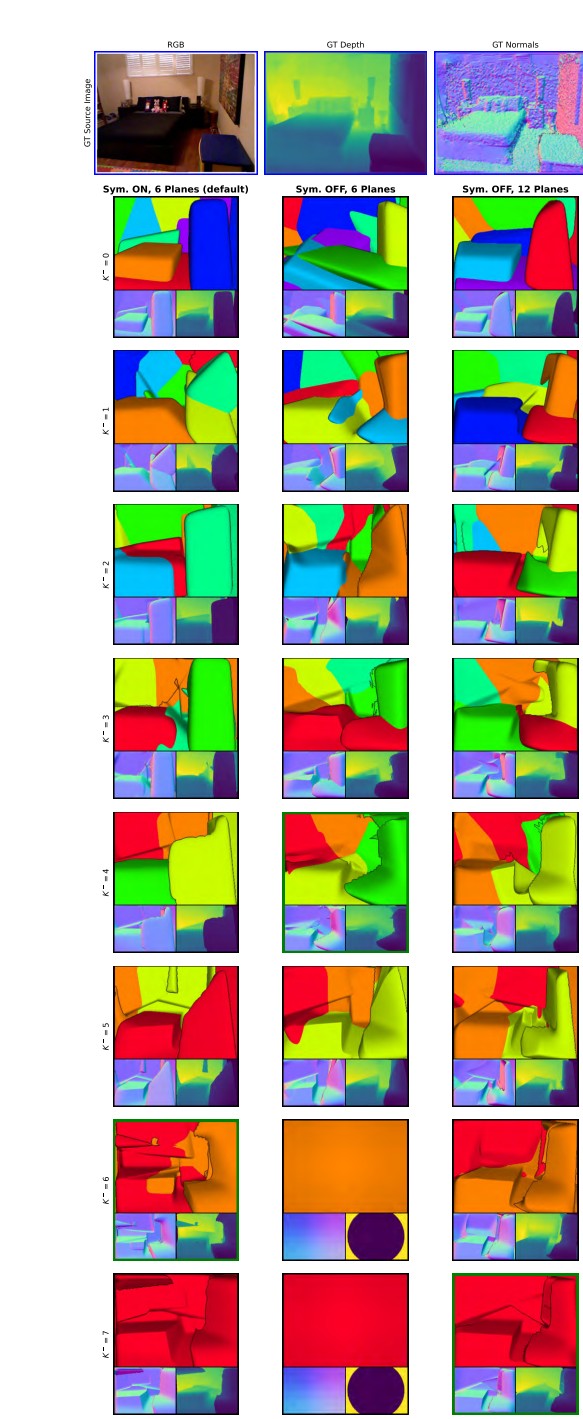

Figure 13: We perform a qualitative evaluation on the number of boolean primitives, $K^- \in [0, 1, ...7]$, with all images having the same $K^{total} = 8$. In each column, the decomposition with lowest AbsRel selected by ensembling is boxed in green. We decompose parallelepipeds with a Manhattan World constraint (**first column**), general 6-face polytopes (**second column**), and 12-face polytopes (**third column**). In the most extreme case, there is one positive primitive and 7 negative primitives whereby the boolean primitives carve geometry away from the positive primitive (**final row**). The final two entries of the middle column reached a degenerate state during the optimization process and failed to recover, which further justifies the benefits of ensembling.

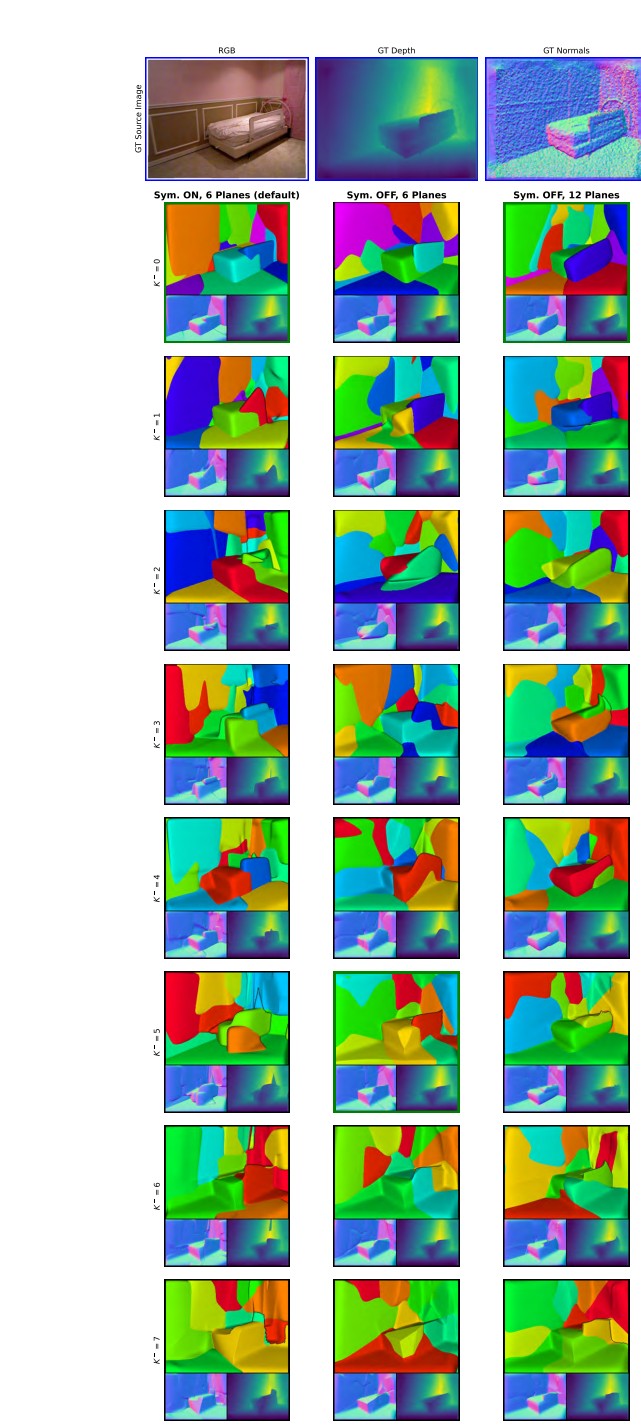

Figure 14: We perform a qualitative evaluation on the number of boolean primitives, $K^- \in [0, 1, ...7]$, with all images having the same $K^{total} = 16$. In each column, the decomposition with lowest AbsRel selected by ensembling is boxed in green. We decompose parallelepipeds with a Manhattan World constraint (**first column**), general 6-face polytopes (**second column**), and 12-face polytopes (**third column**). Notice how the boolean primitives help sharpen the edge of the railing in several cases.

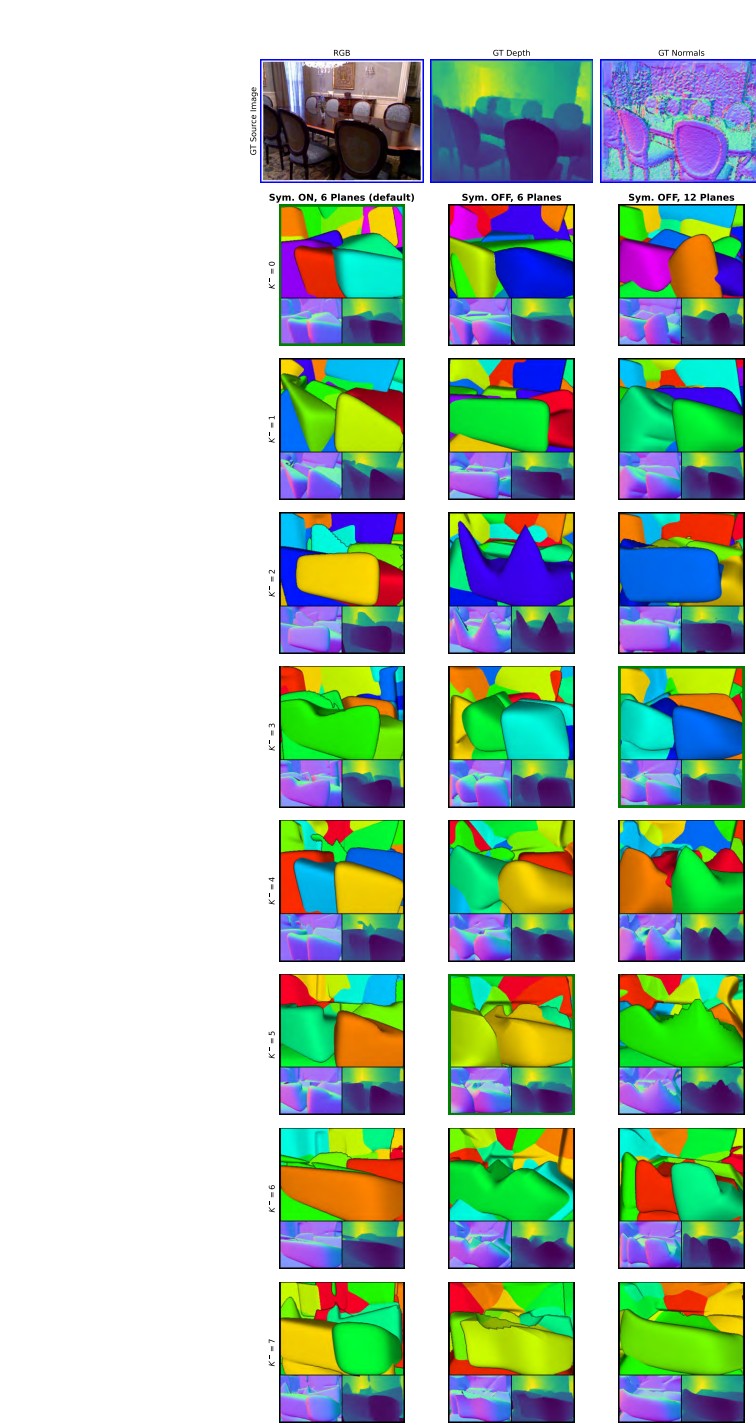

Figure 15: We perform a qualitative evaluation on the number of boolean primitives, $K^- \in [0, 1, ...7]$, with all images having the same $K^{total} = 16$. In each column, the decomposition with lowest AbsRel selected by ensembling is boxed in green. We decompose parallelepipeds with a Manhattan World constraint (**first column**), general 6-face polytopes (**second column**), and 12-face polytopes (**third column**). Notice how the boolean primitives help carve away geometry on the chairs to better model the seat, most evident in the third column, second to last row.

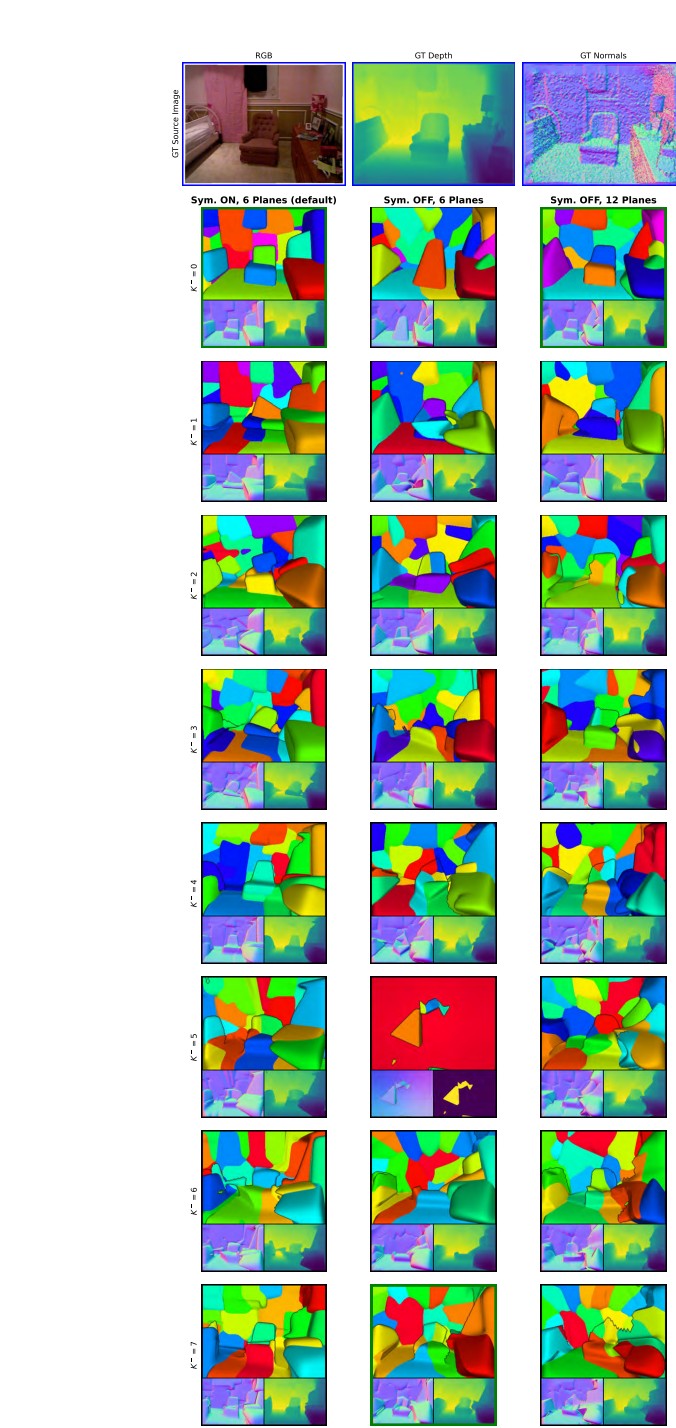

Figure 16: We perform a qualitative evaluation on the number of boolean primitives, $K^- \in [0, 1, ...7]$, with all images having the same $K^{total} = 24$. In each column, the decomposition with lowest AbsRel selected by ensembling is boxed in green. We decompose parallelepipeds with a Manhattan World constraint (**first column**), general 6-face polytopes (**second column**), and 12-face polytopes (**third column**). Notice how the boolean primitives help carve away geometry on the chair and floor.

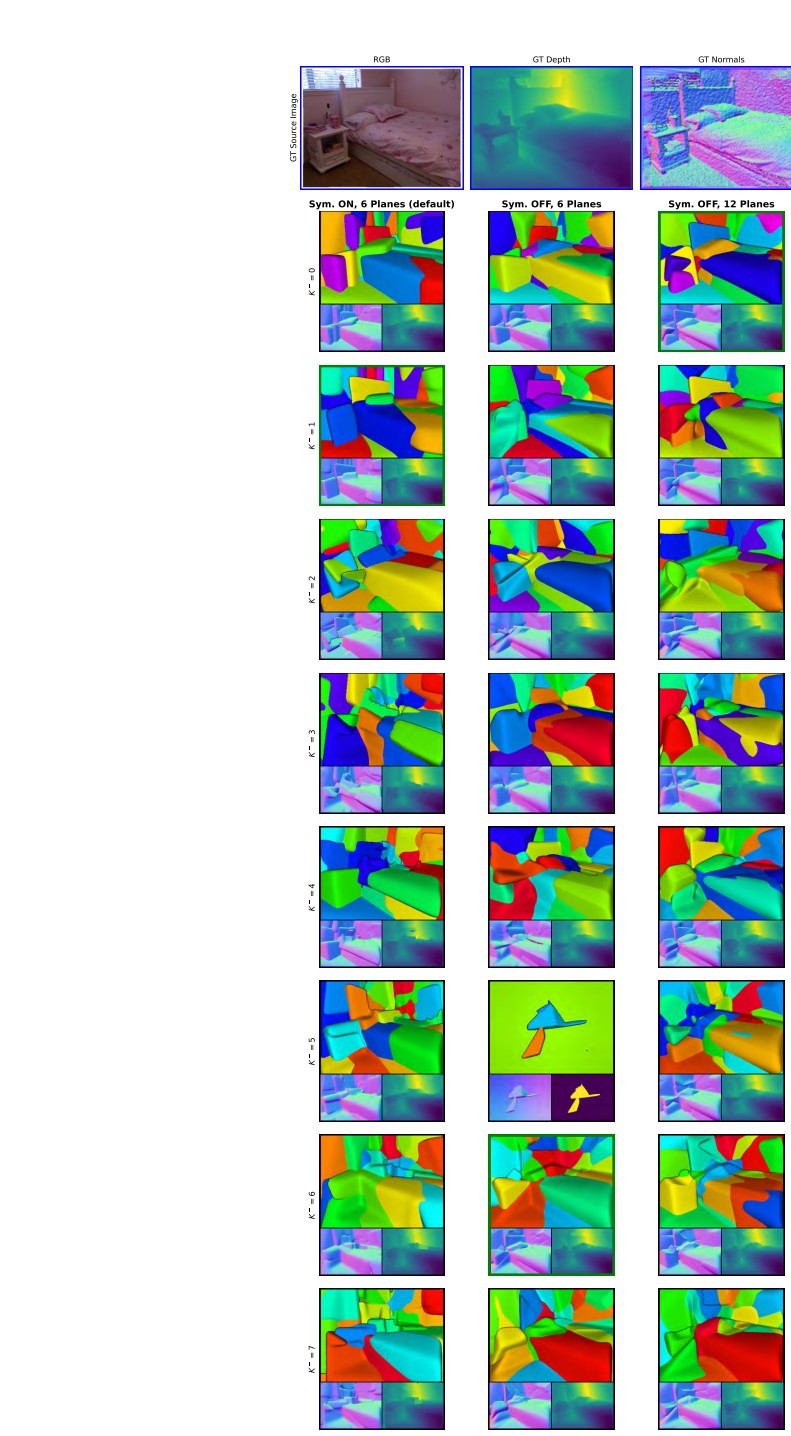

Figure 17: We perform a qualitative evaluation on the number of boolean primitives, $K^- \in [0, 1, ...7]$, with all images having the same $K^{total} = 24$. In each column, the decomposition with lowest AbsRel selected by ensembling is boxed in green. We decompose parallelepipeds with a Manhattan World constraint (**first column**), general 6-face polytopes (**second column**), and 12-face polytopes (**third column**). Notice how the boolean primitives enhance the details of the bed, pillows, and nightstand.

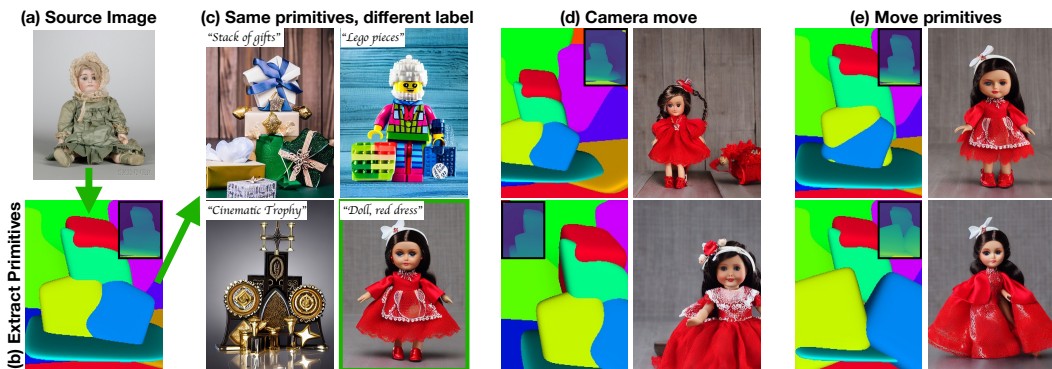

Figure 18: **Our method can decompose natural images into primitives, and be used to condition controlled image synthesis tasks**. We show results from an in-submission follow-up work, which uses the convex decomposition method described here with identical hyperparameters and trains it on a much larger dataset, a 1.8 million-image subset of LAION-Aesthetic. GT depth information was obtained from Yang et al. (2024), and we allow each polytope to use 12 faces without a Manhattan World constraint. We use reasonable camera calibration assumptions to convert the depth map into a point cloud to supervise convex decomposition. We use the same ResNet-18 encoder and 3 FC layer decoder. A validation set reported an AbsRel of 0.130, which is approx. twice the error we report on NYUv2. The larger error on LAION indicates that the images are very diverse and complex in structure as compared with NYUv2. **(a)** We use a convex decomposition method to extract convex polytopes from any image. **(b)** We then ray-march the primitives from the original camera viewpoint to obtain a depth map. **(c)** This depth map serves as conditioning to a ControlNet diffusion model, which is finetuned to handle the unique statistics of our block arrangements. Different scenes can be created from the same high-level geometry. **(d)** We can select one of the images and perform camera moves in 3D space, obtaining images that roughly respect both the requested geometric layout and source texture. We maintain a key-value cache to transfer texture Khachatryan et al. (2023). **(e)** We can also move primitives freely in 3D space, adjusting the high-level shape of the doll's dress.

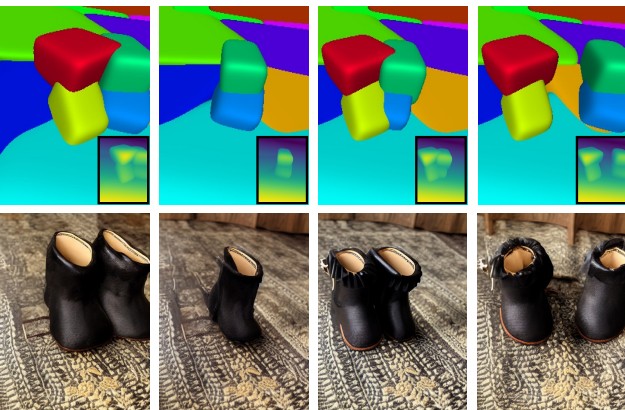

Figure 19: **Our method can decompose natural images into primitives, and be used to condition controlled image synthesis tasks**. We show results from an in-submission follow-up work. Our primitive representation allows us to remove and add objects to a scene, in this case a boot. **Bottom row** We generate an image conditioned on primitives (here, primitives extracted from a real image); we then manipulate the primitives and the camera to obtain conditioning for the diffusion model. Depth and primitives shown in **top row**, generated images in second row. Texture is preserved by caching keys and values from a reference style image, and querying those keys and values when generating new images in the same style.

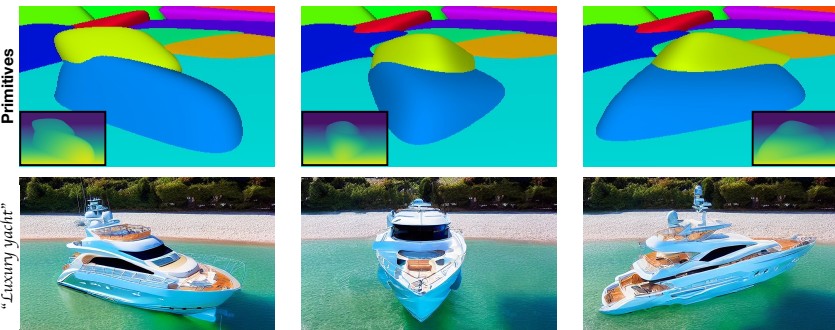

Figure 20: **Our method can decompose natural images into primitives, and be used to condition controlled image synthesis tasks**. We show results from an in-submission follow-up work. Rotating the primitives associated with the yacht rotates the yacht in view.

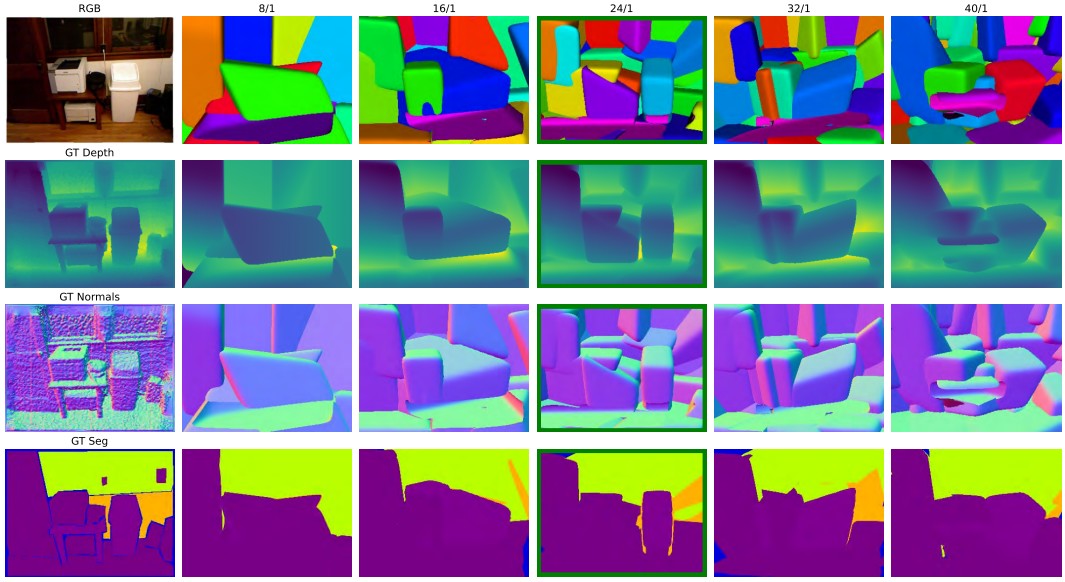

Figure 21: Additional qualitative evaluation with negative primitives. 24/1 was chosen by the ensembling procedure, and the negative primitive was placed on the floor to indicate free space.

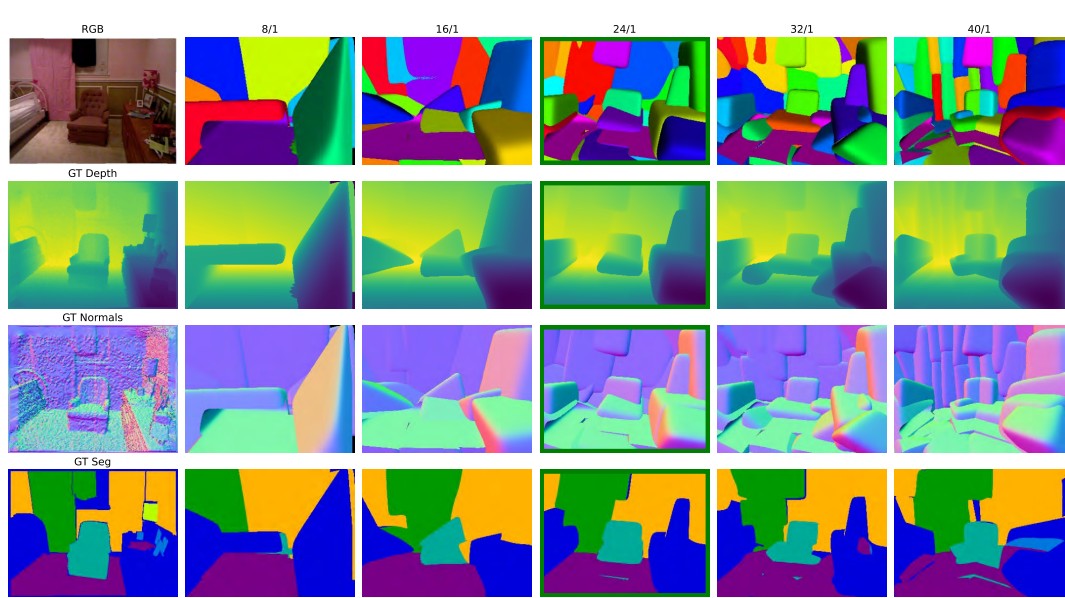

Figure 22: Additional qualitative evaluation with negative primitives. $24/1$ was chosen by the ensembling procedure, and the negative primitive was placed on the floor to indicate free space.

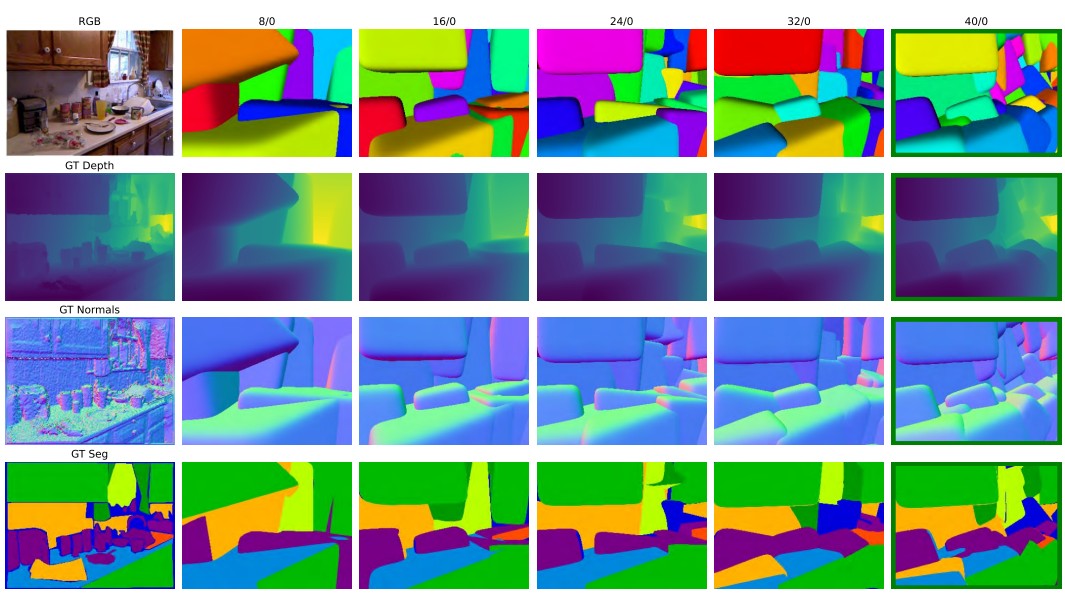

Figure 23: Additional qualitative evaluation with only positive primitives.

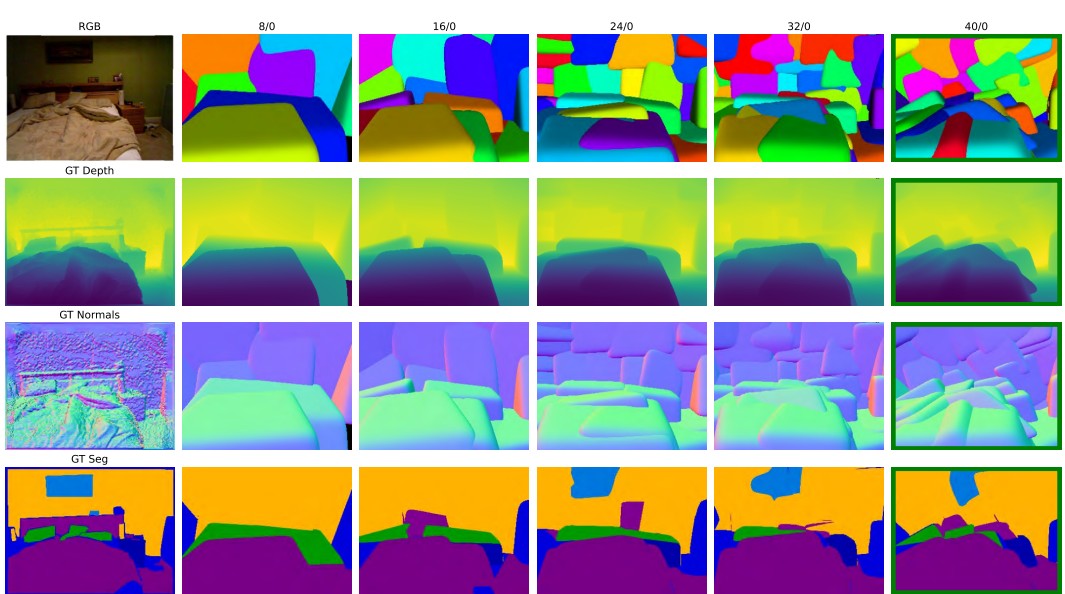

Figure 24: Additional qualitative evaluation with only positive primitives.

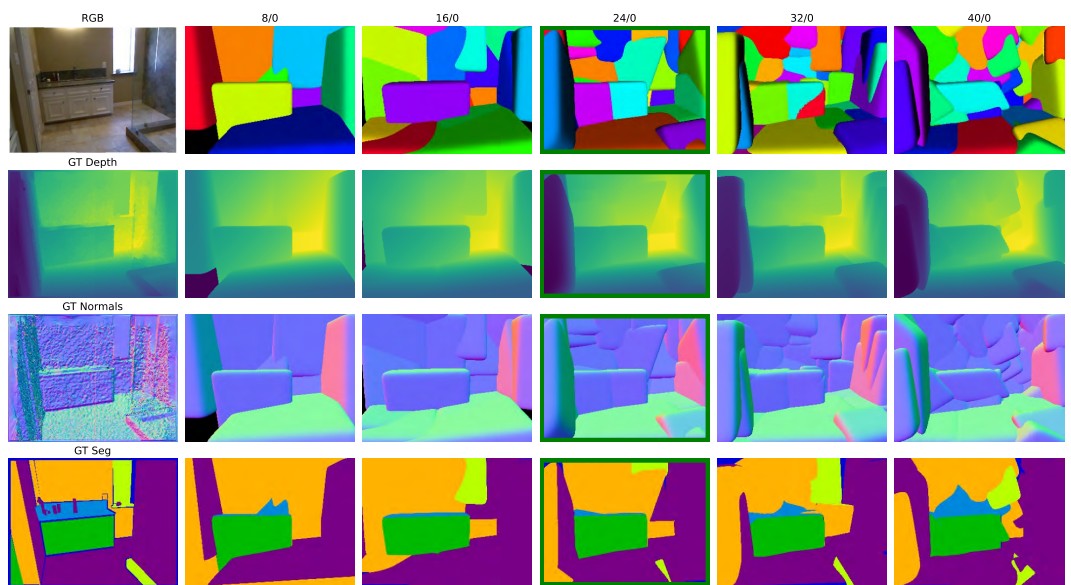

Figure 25: Additional qualitative evaluation with only positive primitives.

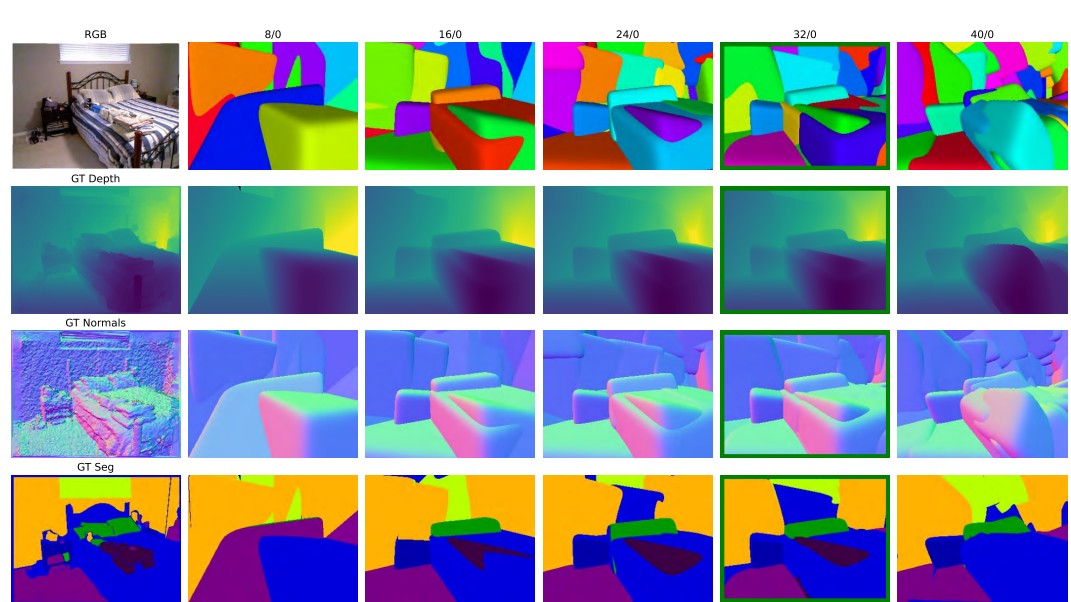

Figure 26: Additional qualitative evaluation with only positive primitives.

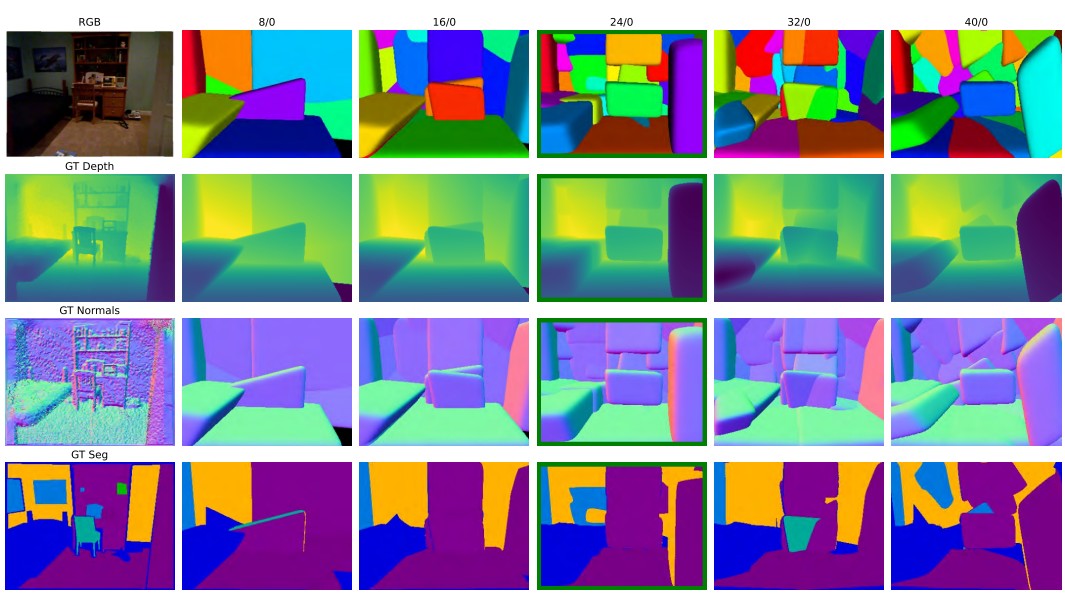

Figure 27: Additional qualitative evaluation with only positive primitives.

