# OpenReview forum: "Improved Convex Decomposition with Ensembling and Boolean Primitives"
_ICLR.cc/2025/Conference — Submitted to ICLR 2025_

### Official Review · Reviewer_o1w1 · 2024-10-28

**Soundness:** 3
**Presentation:** 3
**Contribution:** 3
**Rating:** 6
**Confidence:** 3

**Summary:**

This paper proposes an improved approach to convex decomposition by integrating ensembling techniques and Boolean primitives. The method enhances the accuracy of depth, normal prediction, and segmentation, addressing issues in scene fitting and reducing reliance on a fixed number of primitives by allowing a flexible start point that is iteratively refined.

**Strengths:**

1. Innovative Use of Boolean Primitives: The introduction of negative primitives is a notable addition, enabling more efficient geometric abstraction by allowing finer control over free space representation.

2. Effective Ensembling Strategy: The refine-then-choose approach in ensembling leads to significant improvements in fitting accuracy by allowing multiple regression methods to predict different start points.

**Weaknesses:**

1. High Computational Demand: The ensemble-based approach requires significant training and inference resources, which may limit scalability and real-world applications.

**Questions:**

1. Can the model generalize to more varied datasets given its high resource demands?

---

> ### Author Response · Authors · 2024-11-30
> **Response to questions**
>
> $\textbf{High computational demand}$: indeed, ensembling involves more compute (see Table 1 for time/memory) but can be worth it when quality matters more than cost. Given that fitting primitives is a very difficult problem, having an avenue to overcome poor initialization/fitting will improve robustness. We also point out that for resource-constrained use-cases where ensembling is not an option, we also make several improvements to the underlying fitting procedure such that any individual network we trained (with or without boolean primitives) compares favorably with prior work (see Tables 1 and 2).
>
> $\textbf{Varied datasets}$: We show that our convex decomposition procedure works on general images in Figs. 18-20. We train our model on a subset of LAION (approx. 1.8M varied images) using DepthAnything to obtain a depth map, and reasonable camera calibration assumptions to convert the depth map to a point cloud. Qualitatively, our model abstracts the scene quite well.

---

### Official Review · Reviewer_aCC4 · 2024-10-30

**Soundness:** 2
**Presentation:** 2
**Contribution:** 2
**Rating:** 3
**Confidence:** 4

**Summary:**

This paper studies two techniques to improve the problem of fitting a set of primitives to indoor scenes. Specifically, they first propose to use an ensemble of networks to predict a variety of solutions, and to choose the best one according to certain metrics. They further propose to predict negative primitives to complement with the positive primitives and perform boolean operations, which improves the performances as well. Experiments are conducted to show the effectiveness of the proposed components.

**Strengths:**

- The problem studied in this paper is interesting and important. Decompositing 3D indoor scene geometry into several primitives could facilitate the better understanding of the structure of 3D scene geometry.
- The proposed method in this paper appears to be sound and effective. Performing decomposition with multiple networks will indeed improve the reconstruction result for sure, and adding negative primitives indeed might reduce the representation complexity under certain scenarios.
- Extensive ablation studies are conducted in the paper to show the effectiveness of the proposed compoenents.

**Weaknesses:**

- The writing of this paper is hard to follow. Specifically the paper is based on a prior work, yet the prior work is not fully explained beforehand. Therefore the general method reads disconnected and fails to explain the method well enough.
- It is not clear on the relationship between the proposed method and works that perform CSG decomposition, such as [Du et al. 2018]. However, the negative boolean primitive discussed in this work is relevant to the traditional CSG decompostion of 3D geometry. Moreover, can the proposed method be applied to more general CSG grammar?
- The experiment setting of the evaluation section is not clearly illustrated. Specifically the major metric is the reconstruction metric, which might not convey enough message since the main goal of doing decomposition is to facilitate understanding. Moreover the experiment setting of the proposed method and prior works can be pretty different, since prior works do not predict negative primitives.
- Bringing negative primitives to the prediciton result, although improving reconstruction accuracy, yet can bring other issues. For example, the segmentation and understanding can become more tricky when there are negative primitives.

[Du et al. 2018] InverseCSG: automatic conversion of 3D models to CSG trees.

**Questions:**

- What exactly is the criteria for evaluating a decomposition result? In L195, the authors mention that they use depth map error to decide which decomposition is the best. However, this criteria might not be perfect, since the user might want a decomposition with fewer primitives and clearer structure, despite having larger reconstruction error.

---

> ### Author Response · Authors · 2024-11-30
> **Response to questions**
>
> $\textbf{Explaining prior work}$: It’s challenging to make a paper truly self-contained; we focused the limited space on explaining what’s new and showing results. However, there are only two prior works required to understand what we have done in this work - “CvxNet” and “Convex Decomposition of Indoor Scenes”, and we have made that point in the paper. Further, we have revised the manuscript for clarity and open to improve/explain anything that is confusing.
>
> $\textbf{Comparison with Du 2018}$: The key difference is that we fit implicit surfaces to RGB images in an unsupervised way (we do not know in advance the ground truth primitive decomposition). Du 2018 appears to fit primitives to point clouds where the GT CSG shape program is known in advance. Their method doesn’t involve learning, ours does. Their evaluation tries to match the point cloud from the predicted shape program with the ground truth point cloud. We don’t even know in advance if a fixed primitive vocabulary would apply in our approach; Du 2018 knows in advance that the objects can be represented with CSG trees because that’s how they were made. Thus, we expect our objects to have significant error; they would not.
>
> $\textbf{Additional CSG grammar}$: Our paper already implements union of primitives (inherited concept from CvxNet/Convex Decomposition) and mesh boolean A(x)-B(x) as described in equation (1) of our paper. It’s conceivable to try other operations like intersection, which can be implemented using our indicators e.g. O(x) = A(x)*B(x), where A(x) and B(x) are implicit surfaces composed of a union of primitives.
>
> $\textbf{Evaluation}$: We use the evaluation procedure of “Convex Decomposition of Indoor Scenes 2023” which measure depth, normals, and segmentation accuracy, as well as the error metrics of Kluger 2021. This encompasses both geometric reconstruction quality and object-level alignment of the primitives and the scene. Even though prior works do not use boolean primitives, we still ultimately care about the same things - good geometry and segmentation and therefore our error metrics are reasonable. It is also true that segmentation and scene understanding are more complicated with negative primitives and therefore should be used where we can forego some of the interpretability in exchange for accuracy. Continuing to investigate boolean primitives in the context of scene decomposition is an exciting line of future work - our aim here is to show that they can be fit and improve reconstruction quality in the first place.
>
> $\textbf{Ensembling criteria}$: We use best depth accuracy when picking from the ensemble. You raise a good point that picking the best decomposition should take into account both parsimony and depth accuracy. We think such a metric should be fine-tuned based on the downstream use-case; here our goal is to demonstrate how to get better primitives across a wide range of primitive counts (we show 8-40 in the paper).

---

> > ### Comment · Reviewer_aCC4 · 2024-12-03
> >
> > Thank you for your rebuttal. While the authors' clarification regarding their relationship to CSG decomposition literature is appreciated, several fundamental concerns from the original review remain unaddressed.
> >
> > The authors present two main arguments to justify their evaluation metrics:
> > - The evaluation metrics are inherited from prior work
> > - Their primary application focus is generating controllable depth maps for controllable image diffusion, hence their emphasis on reconstruction metrics
> >
> > However, these arguments are insufficient for the following reasons:
> >
> > First, the adoption of metrics from previous work does not constitute adequate justification for their usage in the current context. Each paper requires careful consideration of appropriate evaluation criteria aligned with specific objectives and claims.
> >
> > Second, the authors state in L34 that their method "should allow simpler, more general reasoning." This positions improved reasoning and understanding as central objectives of primitive-based representations, consistent with the prior works. For instance, CvxNet [1] demonstrates not only reconstruction capabilities but also extensively examines how primitive decomposition enhances abstraction—analogous to the "understanding" and "reasoning" referenced here. The current work would benefit substantially from:
> >
> > - Qualitative comparison of decomposition results against baselines, beyond ablation studies
> > - Deeper analysis of representational interpretability
> > - Quantitative assessment of understanding/reasoning/abstraction capabilities through user studies
> >
> > While the depth-conditioned image generation application is interesting, the paper's scope fundamentally centers on primitive decomposition.
> >
> > Furthermore, the impact of negative primitives on interpretability and abstraction capabilities remains unexplored. When comparing with prior work, such as [2], qualitatively the proposed method demonstrates seemingly worse interpretability in their decomposition. However this is not a side-by-side comparison, so it would be beneficial if we can have one.
> >
> > Overall, the presentation quality of the manuscript does not reach the expected standard for an ICLR paper; it resembles a technical report more closely than a scientific paper. The prose requires refinement, the logical flow is disjointed, and the figures are inadequately prepared—Figure 2 appears to be a raw screenshot from MeshLab if I guess correctly.
> >
> > Given these considerations, the reviewer maintains their original assessment and believes the manuscript currently falls below the bar of ICLR. However, I recognize the research's significant potential and encourage the authors to consider other venues after a major revision.
> >
> > [1] Deng et al., CvxNet: Learnable Convex Decomposition
> > [2] Kluger et al., Cuboids Revisited: Learning Robust 3D Shape Fitting to Single RGB Images

---

> > > ### Author Response · Authors · 2024-12-04
> > >
> > > Thanks for looking at our rebuttal. We respectfully disagree on a few points.
> > >
> > > “the adoption of metrics from previous work does not constitute adequate justification” -> not really right; if everyone invented a new metric for each new paper, we’d have trouble comparing work.
> > >
> > > “comparison … against baselines” - we’d love to.  The only ones are [2] and [3], and we compared to those.
> > >
> > > “CvxNet [1] … extensively examines how primitive decomposition enhances abstraction”  - not really.  CVXNet evaluates: fit vs number of primitives, Fig 9, compare our Tables 1-4; part based retrieval, Fig 10, but not really relevant to indoor scenes; segment labelling accuracy, Fig 11, compare our Tables 1,3,4; baselines, Fig 12, and we did the only ones available for indoor scenes [2] [3]; reconstruction accuracy, Table 1, compare our Tables 1 & 2.
> > >
> > > “Qualitative comparison of decomposition results against baselines” -> please see Fig. 1, which compares against the two available baselines. There is also an enormous amount of qualitative evaluation for different variations of our method in the supplement.
> > >
> > > “Deeper analysis of representational interpretability” - we’re not sure what this means; if it means more than Fig 10 or Fig 11 of CVXNet, then it is likely very hard to achieve in the current state of knowledge.
> > >
> > > “Quantitative assessment of understanding/reasoning/abstraction capabilities through user studies” - user studies of primitive decompositions are a wholly new idea in vision; we aren’t sure how one would do them, and are sure they’re beyond the scope of anything in the area.  Note the user study of [4] looks at control of primitive conditioned image synthesis, so is not really a model.
> > >
> > > “Figure 2 appears to be a raw screenshot from MeshLab if I guess correctly”- This is right; we think the figure is clear, which is what matters.
> > >
> > > [1] Deng et al., CvxNet: Learnable Convex Decomposition
> > > [2] Kluger et al., Cuboids Revisited: Learning Robust 3D Shape Fitting to Single RGB Images
> > > [3] Vavilala et al., Convex Decomposition of Indoor Scenes
> > > [4] Bhat et al., LooseControl: Lifting ControlNet for Generalized Depth Conditioning
> > >
> > >
> > > We want to remind the reviewers that our method produces SOTA results on the established benchmarks by a wide margin, and is supported by extensive quantitative and qualitative evaluation. Further, we’re unaware of any other paper investigating ensembling or boolean primitives in the context of primitive decomposition from RGB images.

---

### Official Review · Reviewer_xiuc · 2024-11-01

**Soundness:** 3
**Presentation:** 3
**Contribution:** 3
**Rating:** 5
**Confidence:** 4

**Summary:**

This paper presents a shape abstraction method using convex primitives for indoor scene understanding. The core contribution lies in the observation that while negative (in the CSG sense) primitives alone provide limited benefits, their combination with model ensembling yields notable performance gains. The method is evaluated on the NYU v2 dataset, using reconstruction accuracy metrics for depth and normal, and segmentation accuracy, following the experimental setups of previous works. Notably, the proposed method significantly outperforms the most relevant SOTA method in depth accuracy.

**Strengths:**

- The paper is well-written and easy to follow.
- The ensemble of results to avoid local optima, though straightforward in general, is novel in learning-based shape abstraction. This ensembling and "pick-best" strategy is well-suited to the setup, where multiple inference results can be evaluated against the given depth map. Interestingly, using negative primitives alone shows limited performance gain, and the paper empirically demonstrates that a refine-then-choose strategy is more effective than the reverse.
- The proposed method outperforms previous works across all metrics, with a notable improvement in depth accuracy over the current SOTA.
- The bias loss term is well-motivated for negative primitives.

**Weaknesses:**

- According to the descriptions of Figures 10 and 11, the negative primitive is used only for the floor. This raises concerns about whether the negative primitive is functioning as intended, as indicated in Figure 2.
- While following the experimental setup of previous works, the proposed method is only evaluated on the NYU v2 dataset, which is relatively small compared to other commonly used benchmarks for indoor RGB-D data, like SUN-RGBD and ScanNet. Evaluation on these larger datasets would strengthen the paper significantly.
- Although well-motivated, the impact of the bias loss term does not seem significant or consistently effective in improving accuracy, as shown in Figure 8.
- An ablation study on learning rate annealing is missing.
- The paper lacks a theoretical explanation on why using negative primitives alone provides limited benefits and combining them with the ensemble method yields better performance.

**Questions:**

I wonder how each proposed component makes a qualitative difference.

---

> ### Author Response · Authors · 2024-11-30
> **Response to questions**
>
> $\textbf{Use of negative primitives}$: We show additional qualitative behavior of the boolean primitives in Figs. 12-17, where they help model bookshelves, chairs, and beds.
>
> $\textbf{Other datasets}$: We show our method works on a larger-scale subset of LAION in Figs. 18-20, reporting an AbsRel of 0.130 when using 12 positive primitives with 12 faces each.
>
> $\textbf{Ablation on learning rate decay}$: Please see Table 5.
>
> $\textbf{Use of bias term}$: Fig. 8 shows that having a small amount of the bias term is generally helpful when fitting boolean primitives, but indeed it’s not a consistent benefit in every case.
>
> $\textbf{Theoretical explanation of negative primitives}$: We’re unaware of theoretical methods to model the performance of boolean primitives; however in Fig. 11 we provide an experimental argument showing that fitting boolean parallelepipeds is very difficult, but relaxing the symmetry constraint and Manhattan world loss significantly improves the fitting of boolean primitives.

---

### Official Review · Reviewer_3wC2 · 2024-11-02

**Soundness:** 3
**Presentation:** 2
**Contribution:** 1
**Rating:** 5
**Confidence:** 4

**Summary:**

This paper presents an approach for decomposing indoor scenes into convex primitives from a single RGBD image. It introduces two key strategies for convex decomposition: the ensembling process and the use of boolean primitives. The ensembling strategy involves employing multiple networks, each predicting a different number or type of primitives, and selecting the optimal primitive set based on refinement loss. The boolean primitives strategy utilizes constructive solid geometry (CSG) operations, allowing up to two negative primitives for subtraction, which enhances geometric accuracy by trimming unnecessary parts of positive primitives. Additionally, the paper proposes skills to improve fitting accuracy, such as biasing sample loss, annealing loss weights, and data augmentation. Experiments on the NYUv2 dataset demonstrate that the proposed strategies enhance geometric representation accuracy and outperform previous state-of-the-art methods in depth estimation, normal prediction, and scene segmentation accuracy. While the method proposed in the paper enhances task performance, some design aspects remain somewhat puzzling and could be further improved.

**Strengths:**

1.This paper tackles a challenging and unresolved task of scene convex decomposition from a single RGBD image and significantly advances the performance on the NYUv2 dataset.
2.It is reasonable to introduce negative primitives as a subtraction operator in CSG, which adds flexibility to primitive-based scene representations. The authors show examples demonstrating that boolean primitives are parameter-efficient for fitting certain geometric shapes. Additionally, the ensembling strategy is straightforward and easy to follow.
3.The paper includes extensive ablation experiments to validate the contribution and performance improvement of each proposed module.

**Weaknesses:**

1.The main contributions of this paper are ensembling predictions from multiple networks and introducing negative primitives. However, these strategies do not appear to yield consistent performance gains. In Table 1, the naive method '24/32/40' outperforms both 'Pos - S→R' and 'Pos + Neg S→R', suggesting that ensembling and negative primitives are not accurate enough to help candidate selection before refinement. Although 'Pos - R→S' and 'Pos+Neg - R→S' achieve better performance, this is likely due to the availability of more final results for selection, indicating that the "refinement then selection"  is still necessary and impactful. In Table 2, it is also somewhat puzzling that, in some cases, introducing negative primitives reduces fitting accuracy.
2.The ensembling strategy is simple but underdeveloped. While it provides more candidate sets of convexes, it also increases computation time significantly. Besides, the number of primitives is preset, meaning the approach is only effective if sufficient networks are available to generate adequate candidates across diverse scenarios. A more effective approach might be to fuse all convexes from multiple networks and then learn to select the best results.
3.Boolean primitives convincingly provide a more accurate and efficient scene representation. However, it is confusing that the maximum number of negative primitives is limited to only 2, and in some cases, the negatives actually reduce performance, as shown in Table 2. Additionally, the paper notes that the primary performance gains from the negative operation occur when a negative primitive occupies empty floor space, whereas it seems more reasonable that negative primitives should have greater potential to remove excess space occupied by positive primitives.
4.In the experiments, the paper mainly compares its results to "Convex decomposition of indoor scenes," but lacks a comparison to "Robust Shape Fitting for 3D Scene Abstraction, IEEE Transactions on Pattern Analysis and Machine Intelligence, 2024."  Besides, while shape abstraction is a lighter-weight representation, it would be appreciated to include application examples of convex decomposition for completeness.

**Questions:**

See above.

---

> ### Author Response · Authors · 2024-11-30
> **Response to questions**
>
> $\textbf{Ensembling selection}$: Indeed our experiments suggest that selecting then refining (pos/pos +neg S->R Table 4) obtains slightly worse AbsRel than 24/32/40 pos only without ensembling. However, the ensembles use 23.3 and 24.5 primitives respectively on average, and we know that more primitives is generally better (up to about 40 primitives). Further, noting that R->S performs substantially better than S->R would indicate that the best start point doesn’t necessarily yield the best end point.
>
> $\textbf{Negative primitives reducing fitting accuracy}$:
> Boolean primitives change the range of geometries we can encode, but they give search problems, as indicated by some scenes showing worse error metrics when replacing positive primitives with negatives. By using boolean primitives, we can encode any scene better if we can find the right fit. But the wide range of geometries we can encode results in variance problems. So in some cases we get better, in other cases we get worse. That’s why we ensemble.
>
> $\textbf{Ensembling and fusing predictions}$:
> What does it mean to fuse all convexes from multiple networks? It was unclear to us how to do that. We agree that our ensembling strategy is simple; we kept it simple because we don’t see how to solve problems created by more complicated strategies. Fusing multiple networks might be much harder than it sounds. We’re looking at a primitive soup of 360 primitives if we consider all 15 members of the ensemble. It’s not clear how to select from this primitive soup. What we have is a selection procedure that convincingly exploits the critical feature of this problem: we can estimate how we’re doing on test data.
>
> $\textbf{Comparison with “Robust Shape Fitting 2024”}$: we compare with the conference version of this paper (which shows identical error metrics) in Table 2.
>
> $\textbf{Applications of convex decomposition}$: Please see Figs. 18-20.

---

### Author Response · Authors · 2024-11-30
**New manuscript uploaded**

We thank the reviewers for their time reading our paper and providing valuable feedback to improve our work. We have uploaded a revised manuscript. In addition to fixing minor typos/improving some explanations, Table 5 (learning rate decay ablation) is new. We add an ablation on the role of network-initialization for post-training finetuning in Fig. 9. Figures 11-17 examine the optimal number of boolean primitives. Our pre-rebuttal submission only investigated up to 2 boolean primitives, because as we show, the quality gets worse the more boolean primitives we use. However, we demonstrate in Fig. 11 that if we relax the parallelepiped assumption and fit general convex polytopes, then more negative primitives are actually helpful. Our quantitative evaluation in the main text uses parallelepipeds, which makes for a fair evaluation against prior work which also uses parallelepipeds/cuboids. But given that our representation is flexible enough to support general convex polytopes, we show that doing so yields very effective boolean primitive-fitting. In Figs. 12-17, the first column of images fits parallelepipeds; the second and third columns fit 6 face and 12 face convex polytopes respectively. In Figs. 18-20, we show that our method can fit primitives to varied general images and isn’t limited to indoor NYUv2 scenes.

We’re happy to engage in further discussion and clarify anything that is confusing - looking forward to hearing additional feedback!

---

### Author Response · Authors · 2024-12-02
**Please acknowledge you've had a chance to see our responses to your questions**

Thanks again for looking at our work!

---

### Meta-Review · Area_Chair_PLHZ · 2024-12-18

**Metareview:**

This paper introduces ideas to improve a previous method for reconstructing an indoor scene into a set of primitives from an RGBD image. Building upon the method by Vavilala and Forsyth, the authors propose two main ideas: 1) incorporating negative primitives, and 2) ensembling multiple regressors. The experiments demonstrate the improvement of these two ideas compared to the baseline method.

Most reviewers gave negative scores and raised concerns about the lack of explanation and justification for why negative primitives and ensembling improve the results, as well as the lack of qualitative analysis. The post-rebuttal discussion quickly converged to reject the submission, and the AC agrees with the reviewers' comments.

**Additional Comments On Reviewer Discussion:**

Please see the Metareview.

---

### Decision · Program_Chairs · 2025-01-22

Reject